# A Theory of Link Prediction via Relational Weisfeiler-Leman on Knowledge Graphs

**Xingyue Huang**
Department of Computer Science
University of Oxford
Oxford, UK.
xingyue.huang@cs.ox.ac.uk

**Miguel Romero**
Department of Computer Science
Universidad Católica de Chile
& CENIA Chile
mgromero@uc.cl

**İsmail İlkan Ceylan**
Department of Computer Science
University of Oxford
Oxford, UK.
ismail.ceylan@cs.ox.ac.uk

**Pablo Barceló**
Inst. for Math. and Comp. Eng.
Universidad Católica de Chile
& IMFD Chile & CENIA Chile
pbarcelo@uc.cl

## Abstract

Graph neural networks are prominent models for representation learning over graph-structured data. While the capabilities and limitations of these models are well-understood for simple graphs, our understanding remains incomplete in the context of knowledge graphs. Our goal is to provide a systematic understanding of the landscape of graph neural networks for knowledge graphs pertaining to the prominent task of link prediction. Our analysis entails a unifying perspective on seemingly unrelated models and unlocks a series of other models. The expressive power of various models is characterized via a corresponding relational Weisfeiler-Leman algorithm. This analysis is extended to provide a precise logical characterization of the class of functions captured by a class of graph neural networks. The theoretical findings presented in this paper explain the benefits of some widely employed practical design choices, which are validated empirically.

## 1 Introduction

Graph neural networks (GNNs) [27, 12] are prominent models for representation learning over graph-structured data, where the idea is to iteratively compute vector representations of nodes of an input graph through a series of invariant (resp., equivariant) transformations. While the landscape of GNNs is overwhelmingly rich, the vast majority of such models are instances of *message passing neural networks* [11] which are well-studied, leading to a theoretical understanding of their capabilities and limitations [34, 21]. In turn, our understanding is rather limited for GNN models dedicated to learning over *knowledge graphs*, which are applied in a wide range of domains.

To make our context precise, we first consider an extension of message passing neural networks with relation-specific message functions, which we call *relational* message passing neural networks. Two prominent examples of this framework are RGCN [28] and CompGCN [32], and their expressive power has recently been characterized through a dedicated relational Weisfeiler-Leman test [4].

While offering principled means for learning over knowledge graphs, the standard relational message passing framework is tailored for computing *unary* node representations and therefore models of this class are better suited for node-level tasks (e.g., node/entity classification). Actually, it is well-known that even a good node-level representation might not necessarily induce a good edge representation,

37th Conference on Neural Information Processing Systems (NeurIPS 2023).

hindering the applicability of such models for the crucial task of link prediction [39]. This has led to the design of GNN architectures specifically tailored for link prediction over knowledge graphs [30, 40, 33, 18], for which our understanding remains limited.

The goal of this paper is to offer a theory of the capabilities and limitations of a class of relational GNN architectures which compute *pairwise* node representations to be utilized for link prediction. Although most such architectures can be seen to be subsumed by *higher-order* message passing neural networks that compute pairwise representations on nodes, the inherently quadratic behavior of the latter justifies local approximations which align better with models used in practice. Of particular interest to us is Neural Bellman-Ford Networks (NBFNets) [40], which define a message passing approach inspired by the Bellman-Ford algorithm. We argue in salient detail that the crucial insight of this approach is in leveraging the idea of computing *conditional* pairwise-node representations, which leads to more expressive models at a more reasonable computational cost, given its local nature.

Building on this fundamental aspect, we define *conditional* message passing neural networks, which extend traditional ones by a conditional message passing paradigm: every node representation is conditional on a source node and a query relation, which allows for computing pairwise node representations. This framework strictly contains NBFNets and allows for a systematic treatment of various other models. Through a careful study of this framework, we can explain the conceptual differences between different models along with their respective expressive power.

Our contributions can be summarized as follows:

- We introduce *conditional* message passing neural networks which encode representations of nodes $v$ conditioned on a (source) node $u$ and a query relation $q$, yielding pairwise node representations. We discuss the model design space, including a discussion on different initialization regimes and the presence of global readout functions in each layer of the network.

- We define a relational Weisfeiler-Leman algorithm (building on similar works such as Barceló et al. [4]), and prove that *conditional* message passing neural networks can match the expressive power of this algorithm. This study reveals interesting insights about NBFNets, suggesting that their strong empirical performance is precisely due to the expressive power, which can be matched by other instances of this framework.

- Viewing *conditional* message passing neural networks (with or without global readouts) as classifiers over pairs of nodes, we give logical characterizations of their expressive power based on formulas definable in some *binary* variants of *graded modal logic* [8, 19]. This provides us with a declarative and well-studied formal counterpart of C-MPNNs.

- We conduct an experimental analysis to verify the impact of various model choices, particularly pertaining to initialization, history, message computation, and global readout functions. We also conduct both inductive and transductive experiments on various real-world datasets, empirically validating our theoretical findings.

## 2 Related work and motivation

Early GNNs for knowledge graphs are relational variations of message passing neural networks. A prototypical example is the RGCN architecture [28], which extends graph convolutional networks (GCNs) [17] with relation-specific message functions. CompGCN [32] and several other architectures [37] follow this line of work with differences in their aggregate, update, and message functions. These architectures encode *unary* node representations and typically rely on a *pairwise decoder* function to predict the likelihood of a link which is known to be suboptimal for link prediction [39]. There is a good understanding of the expressive power of these architectures [4]: we generalize these results in Section 3, since they form the basis for the rest of our results.

A different approach is given for single-relational graphs by SEAL [38], where the idea is to encode (labeled) subgraphs (instead of nodes). GraIL [30] extends this idea to knowledge graphs, and one important virtue of these models is that they are *inductive* even if there are no node features in the input graph. The idea is to use a form of labeling trick [39] based on pairwise shortest path distances in sampled subgraphs, but these architectures suffer from scalability issues. More recent inductive architectures integrate ideas from earlier path-based link prediction approaches [24, 14] into modern GNN architectures, resulting in proposals such as PathCon [33], Geodesic GNNs [18], and NBFNets [40]. Our study is very closely related to NBFNets which is inspired by the generalized

version of the Bellman-Ford algorithm for finding shortest paths. These architectures aggregate over relational paths by keeping track of conditional pairwise-node representations. While NBFNets can be intuitively seen as the neural counterpart of the *generalized* Bellman-Ford algorithm, they do *not* provably align with this algorithm since the "semiring assumption" is invalidated through the use of non-linearities (which is explicit in Zhu et al. [40]). This leaves open many questions regarding the capabilities and limitations of these architectures.

In this paper, we argue that the key insight behind architectures such as NBFNets is in locally computing *pairwise representations through conditioning* on a source node, and this has roots in earlier works, such as ID-GNNs [36]. To formally study this, we introduce conditional message passing neural networks as a strict generalization of NBFNets [40] and related models such as NeuralLP [35], or DRUM [25]. Through this abstraction, we theoretically study the properties of a large class of models in relation to local variants of relational Weisfeiler-Leman algorithms. Broadly, our study can be seen as the relational counterpart of the expressiveness studies conducted for GNNs [34, 21, 3], particularly related to higher-order GNNs [21], which align with higher-order dimensional variants of the WL test. Our characterization relies on *local* versions of higher-order WL tests [22], albeit not in a relational context. This can be seen as a continuation and generalization of the results given for relational message passing neural networks [4] to a broader class of models.

## 3 Background

### 3.1 Knowledge graphs and invariants

**Knowledge graphs.** A *knowledge graph* is a tuple $G = (V, E, R, c)$, where $V$ is a set of nodes, $E \subseteq R \times V \times V$ is a set of labeled edges, or facts, $R$ is the set of relation types and $c : V \to D$ is a node coloring. When $D = \mathbb{R}^d$, we also say that $c$ is a $d$-dimensional *feature map*, and typically use $\boldsymbol{x}$ instead of $c$. We write $r(u, v)$ to denote a labeled edge, or a fact, where $r \in R$ and $u, v \in V$. The *neighborhood* of a node $v \in V$ relative to a relation $r \in R$ is defined as $\mathcal{N}_r(v) := \{u \mid r(u, v) \in E\}$.

**Graph invariants.** We define $k$-*ary graph invariants* following the terminology of Grohe [13], for which we first define isomorphism over knowledge graphs. An *isomorphism* from a knowledge graph $G = (V, E, R, c)$ to a knowledge graph $G' = (V', E', R, c')$ is a bijection $f : V \to V'$ such that $c(v) = c'(f(v))$ for all $v \in V$, and $r(u, v) \in E$ if and only if $r(f(u), f(v)) \in E'$, for all $r \in R$ and $u, v \in V$. A 0-*ary graph invariant* is a function $\xi$ defined on knowledge graphs such that $\xi(G) = \xi(G')$ for all isomorphic knowledge graphs $G$ and $G'$. For $k \geq 1$, a $k$-*ary graph invariant* is a function $\xi$ that associates with each knowledge graph $G = (V, E, R, c)$ a function $\xi(G)$ defined on $V^k$ such that for all knowledge graphs $G$ and $G'$, all isomorphisms $f$ from $G$ to $G'$, and all $k$-tuples of nodes $\boldsymbol{v} \in V^k$, it holds that $\xi(G)(\boldsymbol{v}) = \xi(G')(f(\boldsymbol{v}))$. If $k = 1$, this defines a *node invariant*, or *unary invariant*, and if $k = 2$, this defines a *binary invariant*, which is central to our study.

**Refinements.** A function $\xi(G) : V^k \to D$ *refines* a function $\xi'(G) : V^k \to D$, denoted as $\xi(G) \preceq \xi'(G)$, if for all $\boldsymbol{v}, \boldsymbol{v}' \in V^k$, $\xi(G)(\boldsymbol{v}) = \xi(G)(\boldsymbol{v}')$ implies $\xi'(G)(\boldsymbol{v}) = \xi'(G)(\boldsymbol{v}')$. We call such functions *equivalent*, denoted as $\xi(G) \equiv \xi'(G)$, if $\xi(G) \preceq \xi'(G)$ and $\xi'(G) \preceq \xi(G)$. A $k$-ary graph invariant $\xi$ *refines* a $k$-ary graph invariant $\xi'$, if $\xi(G)$ refines $\xi'(G)$ for all knowledge graphs $G$.

### 3.2 Relational message passing neural networks

We introduce *relational message passing neural networks (R-MPNNs)*, which encompass several known models such as RGCN [28] and CompGCN [32]. The idea is to iteratively update the feature of a node $v$ based on the different relation types $r \in R$ and the features of the corresponding neighbors in $\mathcal{N}_r(v)$. In our most general model we also allow readout functions, that allow further updates to the feature of $v$ by aggregating over the features of all nodes in the graph.

Let $G = (V, E, R, \boldsymbol{x})$ be a knowledge graph, where $\boldsymbol{x}$ is a feature map. An *R-MPNN* computes a sequence of feature maps $\boldsymbol{h}^{(t)} : V \to \mathbb{R}^{d(t)}$, for $t \geq 0$. For simplicity, we write $\boldsymbol{h}_v^{(t)}$ instead of $\boldsymbol{h}^{(t)}(v)$. For each node $v \in V$, the representations $\boldsymbol{h}_v^{(t)}$ are iteratively computed as:

$$\boldsymbol{h}_v^{(0)} = \boldsymbol{x}_v$$
$$\boldsymbol{h}_v^{(t+1)} = \text{UPD}\left(\boldsymbol{h}_v^{(f(t))}, \text{AGG}(\{\!\{\text{MSG}_r(\boldsymbol{h}_w^{(t)}) \mid w \in \mathcal{N}_r(v), r \in R\}\!\}), \text{READ}(\{\!\{\boldsymbol{h}_w^{(t)} \mid w \in V\}\!\})\right),$$

where UPD, AGG, READ, and MSG$_r$ are differentiable *update*, *aggregation*, *global readout*, and relation-specific *message* functions, respectively, and $f : \mathbb{N} \to \mathbb{N}$ is a *history* function[1], which is always non-decreasing and satisfies $f(t) \leq t$. An R-MPNN has a fixed number of layers $T \geq 0$, and then, the final node representations are given by the map $\boldsymbol{h}^{(T)} : V \to \mathbb{R}^{d(T)}$.

The use of a readout component in message passing is well-known [5] but its effect is not well-explored in a relational context. It is of interest to us since it has been shown that standard GNNs can capture a larger class of functions with global readout [3].

An R-MPNN can be viewed as an encoder function enc that associates with each knowledge graph $G$ a function $\mathsf{enc}(G) : V \to \mathbb{R}^{d(T)}$, which defines a node invariant corresponding to $\boldsymbol{h}^{(T)}$. The final representations can be used for node-level predictions. For link-level tasks, we use a binary decoder $\mathsf{dec}_q : \mathbb{R}^{d(T)} \times \mathbb{R}^{d(T)} \to \mathbb{R}$, which produces a score for the likelihood of the fact $q(u, v)$, for $q \in R$.

In Appendix A.1 we provide a useful characterization of the expressive power of R-MPNNs in terms of a relational variant of the Weisfeiler–Leman test [4]. This characterization is essential for the rest of the results that we present in the paper.

## 4 Conditional message passing neural networks

R-MPNNs have serious limitations for the task of link prediction [39], which has led to several proposals that compute pairwise representations directly. In contrast to the case of R-MPNNs, our understanding of these architectures is limited. In this section, we introduce the framework of conditional MPNNs that offers a natural framework for the systematic study of these architectures.

Let $G = (V, E, R, \boldsymbol{x})$ be a knowledge graph, where $\boldsymbol{x}$ is a feature map. A *conditional message passing neural network* (C-MPNN) iteratively computes pairwise representations, relative to a fixed query $q \in R$ and a fixed node $u \in V$, as follows:

$$\boldsymbol{h}_{v|u,q}^{(0)} = \mathrm{INIT}(u, v, q)$$

$$\boldsymbol{h}_{v|u,q}^{(t+1)} = \mathrm{UPD}(\boldsymbol{h}_{v|u,q}^{f(t)}, \mathrm{AGG}(\{\!\!\{ \mathrm{MSG}_r(\boldsymbol{h}_{w|u,q}^{(t)}, \boldsymbol{z}_q) \mid w \in \mathcal{N}_r(v), r \in R \}\!\!\}), \mathrm{READ}(\{\!\!\{ \boldsymbol{h}_{w|u,q}^{(t)} \mid w \in V \}\!\!\})),$$

where INIT, UPD, AGG, READ, and MSG$_r$ are differentiable *initialization*, *update*, *aggregation*, *global readout*, and relation-specific *message* functions, respectively, and $f$ is the history function. We denote by $\boldsymbol{h}_q^{(t)} : V \times V \to \mathbb{R}^{d(t)}$ the function $\boldsymbol{h}_q^{(t)}(u, v) := \boldsymbol{h}_{v|u,q}^{(t)}$, and denote $\boldsymbol{z}_q$ to be a learnable vector representing the query $q \in R$. A C-MPNN has a fixed number of layers $T \geq 0$, and the final pair representations are given by $\boldsymbol{h}_q^{(T)}$. We sometimes write C-MPNNs *without global readout* to refer to the class of models which do not use a readout component.

Intuitively, C-MPNNs condition on a source node $u$ in order to compute representations of $(u, v)$ for all target nodes $v$. To further explain, we show a visualization of C-MPNNs and R-MPNNs in Figure 1 to demonstrate the differences in the forward pass. Contrary to the R-MPNN model where we carry out relational message passing first and rely on the binary decoder to compute a query fact $q(u, v)$, the C-MPNN model first initializes all node representations with the zero vector except the representation of the node $u$ which is assigned a vector with non-zero entry. Following the initialization, we carry out relational message passing and decode the hidden state of the target node $v$ to obtain the output, which yields the representation of $v$ conditioned on $u$.

Observe that C-MPNNs compute binary invariants, provided that the initialization INIT is a binary invariant. To ensure that the resulting model computes pairwise representations, we require $\mathrm{INIT}(u, v, q)$ to be a nontrivial function in the sense that it needs to satisfy *target node distinguishability*: for all $q \in R$ and $v \neq u \in V$, it holds that $\mathrm{INIT}(u, u, q) \neq \mathrm{INIT}(u, v, q)$.

This is very closely related to the *Labeling Trick* proposed by Zhang et al. [39], which is an initialization method aiming to differentiate a set $\{u, v\}$ of target nodes from the remaining nodes in a graph. However, the *Labeling Trick* only applies when both the source $u$ and target nodes $v$ are labeled. Recent state-of-the-art models such as ID-GNN [36] and NBFNet [40] utilize a similar method, but only with the source node $u$ labeled differently in initialization. Our definition captures precisely this, and we offer a theoretical analysis of the capabilities of the aforementioned architectures accordingly.

---

[1]The typical choice is $f(t) = t$, but other options are considered in the literature, as we discuss later.

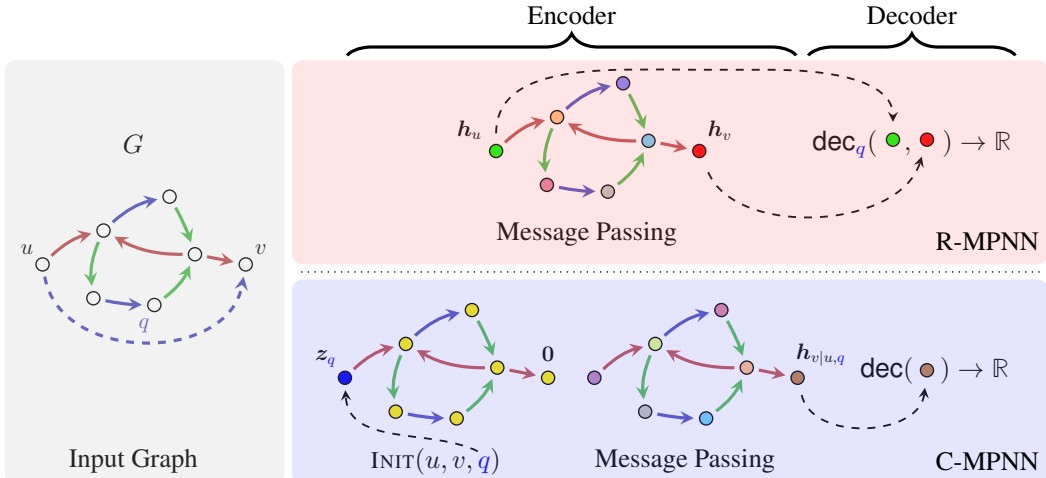

Figure 1: Visualization of R-MPNN and C-MPNN. The dashed arrow is the target query $q(u, v)$. Arrow colors indicate distinct relation types, while node colors indicate varying hidden states. R-MPNN considers a unary encoder and relies on a binary decoder, while C-MPNN first initializes binary representation based on the target query $q(u, v)$, and then uses a unary decoder.

One alternative is to directly learn pairwise representations following similar ideas to those of higher-order GNNs, but these algorithms are not scalable. Architectures such as NBFNets represent a trade-off between computational complexity and expressivity. The advantage of learning conditional representations $\boldsymbol{h}_{v|u,q}$ is to be able to learn such representations in parallel for all $v \in V$, amortizing the computational overhead; see Zhu et al. [40] for a discussion. We have also carried out a runtime analysis comparison among different classes of models in Appendix B.

## 4.1 Design space and basic model architectures

To specify a C-MPNN architecture, we need to specify the functions INIT, AGG, $\text{MSG}_r$, $f$, and READ. In the following, we consider three initialization functions $\{\text{INIT}^1, \text{INIT}^2, \text{INIT}^3\}$, two aggregation functions $\{\text{sum}, \text{PNA}\}$, three message functions $\{\text{MSG}_r^1, \text{MSG}_r^2, \text{MSG}_r^3\}$, two history functions $\{f(t) = t, f(t) = 0\}$, and either *sum global readout* or no readout term.

**Initialization.** We consider the following natural variants for initialization:

$$\text{INIT}^1(u, v, q) = \mathbb{1}_{u=v} * \boldsymbol{1}, \quad \text{INIT}^2(u, v, q) = \mathbb{1}_{u=v} * \boldsymbol{z}_q, \quad \text{INIT}^3(u, v, q) = \mathbb{1}_{u=v} * (\boldsymbol{z}_q + \epsilon_u),$$

where $*$ represents element-wise multiplication, the function $\mathbb{1}_{u=v}(v)$ is the indicator function which returns the all-ones vector $\boldsymbol{1}$ if $u = v$ and the all-zeros vector $\boldsymbol{0}$ otherwise with corresponding size.

Clearly, both $\text{INIT}^1$ and $\text{INIT}^2$ satisfy *target node distinguishability* assumption if we assume $\boldsymbol{z}_q$ has no zero entry, where $\text{INIT}^2$, in addition, allows query-specific initialization to be considered. Suppose we further relax the condition to be *target node distinguishability in expectation*. Then, $\text{INIT}^3$ can also distinguish between each conditioned node $u$ given the same query vector $\boldsymbol{z}_q$ by adding an error vector $\epsilon_u$ sampled from $\mathcal{N}(0, 1)$ to the conditioned node's initialization.

**Aggregation.** We consider sum aggregation and Principal Neighborhood Aggregation (PNA) [7].

**Message.** We consider the following variations of *message* functions:

$$\text{MSG}_r^1(\boldsymbol{h}_{w|u,q}^{(t)}, \boldsymbol{z}_q) = \boldsymbol{h}_{w|u,q}^{(t)} * \boldsymbol{W}_r^{(t)} \boldsymbol{z}_q,$$

$$\text{MSG}_r^2(\boldsymbol{h}_{w|u,q}^{(t)}, \boldsymbol{z}_q) = \boldsymbol{h}_{w|u,q}^{(t)} * \boldsymbol{b}_r^{(t)},$$

$$\text{MSG}_r^3(\boldsymbol{h}_{w|u,q}^{(t)}, \boldsymbol{z}_q) = \boldsymbol{W}_r^{(t)} \boldsymbol{h}_{w|u,q}^{(t)},$$

where $\boldsymbol{W}_r^{(t)}$ are relation-specific transformations. $\text{MSG}_r^1$ computes a query-dependent message, whereas $\text{MSG}_r^2$ and $\text{MSG}_r^3$ are analogous to message functions of CompGCN and RGCN, respectively.

**History.** In addition, we can set $f$, which intuitively is the function that determines the history of node embeddings to be considered. By setting $f(t) = t$, we obtain a standard message-passing algorithm where the update function considers the representation of the node in the previous iteration. We can alternatively set $f(t) = 0$, in which case we obtain (a generalization of) NBFNets.

**Readout.** We consider a standard readout which sums the representations and applies a linear transformation on the resulting representations. Alternatively, we consider the special case, where we omit this component (we discuss a dedicated readout operation in our empirical analysis later).

**A simple architecture.** Consider a *basic* C-MPNN architecture with global readout, which, for a query relation $q \in R$ and a fixed node $u$, updates the representations as:

$$\boldsymbol{h}_{v|u,q}^{(0)} = \mathbb{1}_{u=v} * \boldsymbol{z}_q$$

$$\boldsymbol{h}_{v|u,q}^{(t+1)} = \sigma\left(\boldsymbol{W}_0^{(t)}\left(\boldsymbol{h}_{v|u,q}^{(t)} + \sum_{r \in R}\sum_{w \in \mathcal{N}_r(v)} \text{MSG}_r^1(\boldsymbol{h}_{w|u,q}^{(t)}, \boldsymbol{z}_q)\right) + \boldsymbol{W}_1^{(t)}\sum_{w \in V} \boldsymbol{h}_{w|u,q}^{(t)}\right),$$

where $\boldsymbol{W}_0^{(t)}, \boldsymbol{W}_1^{(t)}$ are linear transformations followed by a non-linearity $\sigma$.

## 5 Characterizing the expressive power

### 5.1 A relational Weisfeiler-Leman characterization

To analyze the expressive power of C-MPNNs, we introduce the *relational asymmetric local* 2-*WL*, denoted by $\text{rawl}_2$. In this case, we work with knowledge graphs of the form $G = (V, E, R, c, \eta)$, where $\eta : V \times V \to D$ is a *pairwise coloring*. We say that $\eta$ satisfies *target node distinguishability* if $\eta(u, u) \neq \eta(u, v)$ for all $u \neq v \in V$. The notions of isomorphism and invariants extend to this context in a natural way. For each $t \geq 0$, we update the coloring as:

$$\text{rawl}_2^{(0)}(u, v) = \eta(u, v),$$

$$\text{rawl}_2^{(t+1)}(u, v) = \tau\left(\text{rawl}_2^{(t)}(u, v), \{\!\{(\text{rawl}_2^{(t)}(u, w), r) \mid w \in \mathcal{N}_r(v), r \in R\}\!\}\right),$$

where $\tau$ injectively maps the above pair to a unique color, which has not been used in previous iterations. Observe that $\text{rawl}_2^{(t)}$ defines a binary invariant, for all $t \geq 0$.

The test is asymmetric: given a pair $(u, v)$, we only look at neighbors of $(u, v)$ obtained by changing the second coordinate of the pair. In contrast, usual versions of (local) $k$-WL are symmetric as neighbors may change any coordinate. Interestingly, this test characterizes the power of C-MPNNs in terms of distinguishing pairs of nodes.

**Theorem 5.1.** *Let $G = (V, E, R, \boldsymbol{x}, \eta)$ be a knowledge graph, where $\boldsymbol{x}$ is a feature map and $\eta$ is a pairwise coloring satisfying target node distinguishability. Let $q \in R$ be any query relation. Then:*

1. *For all C-MPNNs with $T$ layers and initializations* INIT *with* INIT $\equiv \eta$, *and $0 \leq t \leq T$, we have* $\text{rawl}_2^{(t)} \preceq \boldsymbol{h}_q^{(t)}$.

2. *For all $T \geq 0$ and history function $f$, there is a C-MPNN without global readout with $T$ layers and history function $f$ such that for all $0 \leq t \leq T$, we have* $\text{rawl}_2^{(t)} \equiv \boldsymbol{h}_q^{(t)}$.

The idea of the proof is as follows: we first show a correspondent characterization of the expressive power of R-MPNNs in terms of a relational variant of the WL test (Theorem A.1). This result generalizes results from Barceló et al. [4]. We then apply a reduction from C-MPNNs to R-MPNNs, that is, we carefully build an auxiliary knowledge graph (encoding the pairs of nodes of the original knowledge graph) to transfer our R-MPNN characterization to our sought C-MPNN characterization.

Note that the lower bound (item (2)) holds even for the *basic* model of C-MPNNs (without global readout) and the three proposed message functions from Section 4.1. The expressive power of C-MPNNs is independent of the history function as in any case it is matched by $\text{rawl}_2$. This suggests that the difference between traditional message passing models using functions $f(t) = t$ and path-based models (such as NBFNets [40]) using $f(t) = 0$ is not relevant from a theoretical point of view.

## 5.2 Logical characterization

We now turn to the problem of which *binary classifiers* can be expressed as C-MPNNs. That is, we look at C-MPNNs that classify each pair of nodes in a knowledge graph as true or false. Following Barceló et al. [3], we study *logical* binary classifiers, i.e., those that can be defined in the formalism of first-order logic (FO). Briefly, a first-order formula $\phi(x, y)$ with two free variables $x, y$ defines a logical binary classifier that assigns value true to the pair $(u, v)$ of nodes in knowledge graph $G$ whenever $G \models \phi(u, v)$, i.e., $\phi$ holds in $G$ when $x$ is interpreted as $u$ and $y$ as $v$. A logical classifier $\phi(x, y)$ is *captured* by a C-MPNN $\mathcal{A}$ if over every knowledge graph $G$ the pairs $(u, v)$ of nodes that are classified as true by $\phi$ and $\mathcal{A}$ are the same.

A natural problem then is to understand what are the logical classifiers captured by C-MPNNs. Fix a set of relation types $R$ and a set of pair colors $\mathcal{C}$. We consider knowledge graphs of the form $G = (V, E, R, \eta)$ where $\eta$ is a mapping assigning colors from $\mathcal{C}$ to pairs of nodes from $V$. In this context, FO formulas can refer to the different relation types in $R$ and the different pair colors in $\mathcal{C}$. Our first characterization is established in terms of a simple fragment of FO, which we call $\mathsf{rFO}^3_{\mathrm{cnt}}$, and is inductively defined as follows: First, $a(x, y)$ for $a \in \mathcal{C}$, is in $\mathsf{rFO}^3_{\mathrm{cnt}}$. Second, if $\varphi(x, y)$ and $\psi(x, y)$ are in $\mathsf{rFO}^3_{\mathrm{cnt}}$, $N \geq 1$ is a positive integer, and $r \in R$, then the formulas

$$\neg\varphi(x, y), \quad \varphi(x, y) \wedge \psi(x, y), \quad \exists^{\geq N} z \, (\varphi(x, z) \wedge r(z, y))$$

are also in $\mathsf{rFO}^3_{\mathrm{cnt}}$. Intuitively, $a(u, v)$ holds in $G = (V, E, R, \eta)$ if $\eta(u, v) = a$, and $\exists^{\geq N} z \, (\varphi(u, z) \wedge r(z, v))$ holds in $G$ if $v$ has at least $N$ incoming edges labeled $r \in R$ from nodes $w$ for which $\varphi(u, w)$ holds in $G$. We use the acronym $\mathsf{rFO}^3_{\mathrm{cnt}}$ as this logic corresponds to a restriction of FO with three variables and counting. We can show the following result which is the first of its kind in the context of knowledge graphs:

**Theorem 5.2.** *A logical binary classifier is captured by C-MPNNs without global readout if and only if it can be expressed in* $\mathsf{rFO}^3_{\mathrm{cnt}}$.

The idea of the proof is to show a logical characterization for R-MPNNs (without global readout) in terms of a variant of graded modal logic called $\mathsf{rFO}^2_{\mathrm{cnt}}$ (Theorem A.11), which generalizes results from Barceló et al. [3] to the case of multiple relations. Then, as in the case of Theorem 5.1, we apply a reduction from C-MPNNs to R-MPNNs (without global readout) using an auxiliary knowledge graph and a useful translation between the logics $\mathsf{rFO}^2_{\mathrm{cnt}}$ and $\mathsf{rFO}^3_{\mathrm{cnt}}$.

Interestingly, arbitrary C-MPNNs (with global readout) are strictly more powerful than C-MPNNs without global readout in capturing logical binary classifiers: they can at least capture a strict extension of $\mathsf{rFO}^3_{\mathrm{cnt}}$, denoted by $\mathsf{erFO}^3_{\mathrm{cnt}}$.

**Theorem 5.3.** *Each logical binary classifier expressible in* $\mathsf{erFO}^3_{\mathrm{cnt}}$ *can be captured by a C-MPNN.*

Intuitively speaking, our logic $\mathsf{rFO}^3_{\mathrm{cnt}}$ from Theorem 5.2 only allows us to navigate the graph by moving to neighbors of the "current node". The logic $\mathsf{erFO}^3_{\mathrm{cnt}}$ is a simple extension that allows us to move also to non-neighbors (Proposition A.15 shows that this logic actually gives us more power). Adapting the translation from logic to GNNs (from Theorem A.11), we can easily show that C-MPNNs with global readout can capture this extended logic.

The precise definition of $\mathsf{erFO}^3_{\mathrm{cnt}}$ together with the proofs of Theorems 5.2 and 5.3 can be found in Appendices A.3 and A.4, respectively. Let us stress that $\mathsf{rFO}^3_{\mathrm{cnt}}$ and $\mathsf{erFO}^3_{\mathrm{cnt}}$ correspond to some binary variants of *graded modal logic* [8, 19]. As in Barceló et al. [3], these connections are exploited in our proofs.

## 5.3 Locating $\mathsf{rawl}_2$ in the relational WL landscape

Let us note that $\mathsf{rawl}_2$ strictly contains $\mathsf{rwl}_1$, since intuitively, we can degrade the $\mathsf{rawl}_2$ test to compute unary invariants such that it coincides with $\mathsf{rwl}_1$. As a result, this allows us to conclude that R-MPNNs are less powerful than C-MPNNs. The $\mathsf{rawl}_2$ test itself is upper bounded by a known relational variant of 2-WL, namely, the *relational (symmetric) local 2-WL* test, denoted by $\mathsf{rwl}_2$. Given a knowledge

graph $G = (V, E, R, c, \eta)$, this test assigns pairwise colors via the following update rule:

$$\mathsf{rwl}_2^{(t+1)}(u,v) = \tau\big(\mathsf{rwl}_2^{(t)}(u,v), \{\!\!\{(\mathsf{rwl}_2^{(t)}(w,v),r) \mid w \in \mathcal{N}_r(u), r \in R)\}\!\!\},$$

$$\{\!\!\{(\mathsf{rwl}_2^{(t)}(u,w),r) \mid w \in \mathcal{N}_r(v), r \in R)\}\!\!\}\big)$$

This test and a corresponding neural architecture, for arbitrary order $k \geq 2$, have been recently studied in Barceló et al. [4] under the name of *multi-relational local k-WL*.

This helps us to locate the test $\mathsf{rawl}_2$ within the broader WL hierarchy, but it does not align perfectly with the practical setup: one common practice in link prediction is to extend knowledge graphs with inverse relations [40, 35, 25, 32] which empirically yields stronger results and hence is used in most practical setups. However, the effect of this choice has never been quantified formally. We formally explain the benefits of this design choice, showing that it leads to provably more powerful models. The idea is to consider tests (and architectures) which are augmented with the inverse edges: we write $\mathsf{rawl}_2^+$ and $\mathsf{rwl}_2^+$ to denote the corresponding augmented tests and prove that it results in more expressive tests (and hence architectures) in each case, respectively. The precise tests and propositions, along with their proofs can be found in Appendix A.5. We present in Figure 2 the resulting expressiveness hierarchy for all these tests.

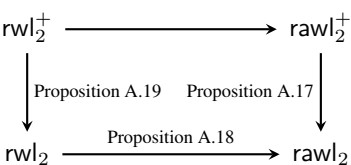

Figure 2: Expressiveness hierarchy: $A \to B$ iff $A \preceq B$. By Proposition A.20, $\mathsf{rwl}_2$ and $\mathsf{rawl}_2^+$ are incomparable. The case of $\mathsf{rwl}_2^+ \preceq \mathsf{rawl}_2^+$ is analogous to Proposition A.18.

# 6 Experimental evaluation

We experiment on knowledge graph benchmarks and aim to answer the following questions: **Q1.** What is the impact of the history function on the model performance? In particular, do models with $f(t) = t$ perform comparably to those with $f(t) = 0$ as our theory suggests? **Q2.** How do the specific choices for aggregation and message functions affect the performance? **Q3.** What is the impact of the initialization function on the performance? What happens when the target identifiability property does not hold? **Q4.** Do C-MPNNs outperform R-MPNNs empirically? **Q5.** Does the use of a global readout, or a relational version affect the performance?

## 6.1 Experimental setup

**Datasets.** We use the datasets WN18RR [31] and FB15k-237 [9], for inductive relation prediction tasks, following a standardized train-test split given in four versions [30]. We augment each fact $r(u,v)$ with an inverse fact $r^{-1}(v,u)$. There are no node features for either of the datasets, and the initialization is given by the respective initialization function INIT. This allows all the proposed GNN models to be applied in the inductive setup and to better align with the corresponding relational Weisfeiler-Leman algorithms. The statistics of the datasets are reported in Table 5 of Appendix C.1. The code for experiments is reported in `https://github.com/HxyScotthuang/CMPNN`.

**Implementation.** All models use 6 layers, each with 32 hidden dimensions. The decoder function parameterizes the probability of a fact $q(u,v)$ as $p(v \mid u,q) = \sigma(f(\boldsymbol{h}_{v|u,q}^{(T)}))$, where $\sigma$ is the sigmoid function, and $f$ is a 2-layer MLP with 64 hidden dimensions. We adopted layer-normalization [2] and short-cut connection after each aggregation and before applying ReLU. For the experiments concerning the message function $\mathsf{MSG}_r^3$, we follow the basis decomposition for the FB15k-237 dataset with 30 basis functions for sum aggregation, and 15 for PNA aggregation. We ran the experiments for 20 epochs on 1 Tesla T4 GPU. with mild modifications to accommodate all architectures studied in this paper. We discard the edges that directly connect query node pairs to prevent overfitting. The best checkpoint for each model instance is selected based on its performance on the validation set. All hyperparameter details are reported in Table 6 of Appendix C.1.

**Evaluation.** We consider *filtered ranking protocol* [6]: for each test fact $r(u,v)$, we construct 50 negative samples $r(u',v')$, randomly replacing either the head entity or the tail entity, and we report Hits@10, the rate of correctly predicted entities appearing in the top 10 entries for each instance list prediction. We report averaged results of *five* independent runs for all experiments.

Table 1: Inductive relation prediction with C-MPNNs using $\text{INIT}^2$ initialization and *no* readout. The best results for each category are shown in **bold** and the second best results are underlined.

| AGG | $\text{MSG}_r$ | $f(t)$ | WN18RR v1 | v2 | v3 | v4 | FB15k-237 v1 | v2 | v3 | v4 |
|---|---|---|---|---|---|---|---|---|---|---|
| sum | $\text{MSG}_r^1$ | $0$ | 0.934 | 0.896 | 0.894 | 0.881 | 0.784 | 0.900 | 0.940 | 0.923 |
| sum | $\text{MSG}_r^1$ | $t$ | 0.932 | 0.896 | **0.900** | 0.881 | 0.794 | 0.906 | **0.947** | 0.933 |
| sum | $\text{MSG}_r^2$ | $0$ | 0.939 | **0.906** | 0.881 | 0.881 | 0.734 | 0.899 | 0.911 | 0.941 |
| sum | $\text{MSG}_r^2$ | $t$ | 0.937 | **0.906** | 0.865 | **0.884** | 0.728 | 0.883 | 0.929 | 0.931 |
| sum | $\text{MSG}_r^3$ | $0$ | **0.943** | 0.898 | 0.888 | 0.877 | **0.850** | 0.934 | 0.919 | 0.941 |
| sum | $\text{MSG}_r^3$ | $t$ | 0.934 | 0.896 | 0.892 | 0.880 | 0.844 | **0.943** | 0.926 | **0.950** |
| PNA | $\text{MSG}_r^1$ | $0$ | 0.943 | 0.897 | 0.898 | 0.886 | 0.801 | 0.945 | 0.934 | **0.960** |
| PNA | $\text{MSG}_r^1$ | $t$ | 0.941 | 0.895 | **0.904** | 0.886 | **0.804** | **0.949** | **0.945** | 0.954 |
| PNA | $\text{MSG}_r^2$ | $0$ | 0.946 | 0.900 | 0.896 | 0.887 | 0.715 | 0.896 | 0.887 | 0.886 |
| PNA | $\text{MSG}_r^2$ | $t$ | **0.947** | **0.902** | 0.901 | **0.888** | 0.709 | 0.899 | 0.875 | 0.894 |
| PNA | $\text{MSG}_r^3$ | $0$ | **0.947** | 0.898 | 0.899 | 0.884 | 0.788 | 0.908 | 0.906 | 0.927 |
| PNA | $\text{MSG}_r^3$ | $t$ | 0.944 | 0.897 | 0.894 | 0.882 | 0.795 | 0.916 | 0.908 | 0.926 |

## 6.2 Results for inductive link prediction with C-MPNN architectures

We report inductive link prediction results for different C-MPNN architectures in Table 1, all initialized with $\text{INIT}^2$. Each row of Table 1 corresponds to a specific architecture, which allows us to compare the model components. Note that while NBFNets [40] use different message functions for different datasets, we separately report for each model architecture to specifically pinpoint the impact of each model component.

**History functions (Q1).** First, we note that there is no significant difference between the models with different history functions. Specifically, for any choice of aggregate and message functions, the model which sets $f(t) = t$ performs comparably to the one which sets $f(t) = 0$. This supports our theoretical findings, which state that path-based message passing and traditional message passing have the same expressive power. This may appear as a subtle point, but it is important for informing future work: the strength of these architectures is fundamentally due to their ability to compute more expressive *binary invariants*, which holds regardless of the choice of the history function.

**Message functions (Q2).** We highlight that there is no significant difference between different message functions on WN18RR, which is unsurprising: WN18RR splits contain at most 11 relation types, which undermines the impact of the differences in message functions. In contrast, the results on FB15k-237 are informative in this respect: $\text{MSG}_r^2$ clearly leads to worse performance than all the other choices, which can be explained by the fact that $\text{MSG}_r^2$ utilizes fewer relation-specific parameters. Importantly, $\text{MSG}_r^3$ appears strong and robust across models. This is essentially the message function of RGCN and uses basis decomposition to regularize the parameter matrices. Architectures using $\text{MSG}_r^3$ with fewer parameters (see the appendix) can match or substantially exceed the performance of the models using $\text{MSG}_r^1$, where the latter is the primary message function used in NBFNets. This may appear counter-intuitive since $\text{MSG}_r^3$ does not have a learnable query vector $z_q$, but this vector is nonetheless part of the model via the initialization function $\text{INIT}^2$.

**Aggregation functions (Q2).** We experimented with aggregation functions $\text{sum}$ and PNA. We do not observe significant trends on WN18RR, but PNA tends to result in slightly better-performing architectures. On FB15k-237, there seems to be an intricate interplay between aggregation and message functions. For $\text{MSG}_r^1$, PNA appears to be a better choice than $\text{sum}$. On the other hand, for both $\text{MSG}_r^2$ and $\text{MSG}_r^3$, sum aggregation is substantially better. This suggests that a sophisticated aggregation, such as PNA, may not always be necessary since it can be matched (and even outperformed) with a sum aggregation. In fact, the model with sum aggregation and $\text{MSG}_r^3$ is very closely related to RGCN and appears to be one of the best-performing models across the board. This supports our theory since, intuitively, this model can be seen as an adaptation of RGCN to compute binary invariants while keeping the choices for model components the same as RGCN.

Table 2: Results for inductive relation prediction on WN18RR and FB15k-237 using *no readout*, *global readout* and *relation-specific readout, respectively*. We use C-MPNN architecture with $\textsc{Agg} = \text{PNA}$, $\textsc{Msg} = \textsc{Msg}_r^1$, and $\textsc{Init} = \textsc{Init}^2$.

| Model architectures | WN18RR | | | | FB15k-237 | | | |
| :---: | :---: | :---: | :---: | :---: | :---: | :---: | :---: | :---: |
| READ | v1 | v2 | v3 | v4 | v1 | v2 | v3 | v4 |
| no readout | 0.941 | **0.895** | **0.904** | **0.886** | 0.804 | 0.949 | 0.945 | 0.954 |
| global readout | **0.946** | 0.890 | 0.881 | 0.872 | 0.799 | 0.897 | 0.942 | 0.864 |
| relation-specific | 0.932 | 0.885 | 0.882 | 0.874 | **0.835** | **0.959** | **0.953** | **0.960** |

We conducted further experiments to understand the differences in different initializations at Appendix C.2, confirming that the initializations that do not satisfy the target node distinguishability criteria perform significantly worse (**Q3**). We could not include R-MPNNs (e.g., RGCN and CompGCN) in these experiments, since these architectures are not applicable to the fully inductive setup. However, we conducted experiments on transductive link prediction in Appendix D.1, and carried out additional experiments on biomedical knowledge in Appendix D.2. In both experiments, we observed that C-MPNNs outperform R-MPNNs by a significant margin (**Q4**).

### 6.3 Empirically evaluating the effect of readout

To understand the effect of readout (**Q5**), we experiment with C-MPNNs using $\textsc{Init}^2$, PNA, $\textsc{Msg}_r^1$, and $f(t) = t$. We consider three architectures: (i) no readout component, (ii) global readout component (as defined earlier), and finally (iii) relation-specific readout. The idea behind the relation-specific readout is natural: instead of summing over all node representations, we sum over the features of nodes $w$ which have an incoming $q$-edge ($w \in V_q^+$) or an outgoing $q$-edge ($w \in V_q^-$), where $q$ is the query relation, as:

$$\textsc{Read}(\{\!\{\boldsymbol{h}_{w|u,q}^{(t)} \mid w \in V_q^+\}\!\}, \{\!\{\boldsymbol{h}_{w|u,q}^{(t)} \mid w \in V_q^-\}\!\}) = \mathbf{W}_1^{(t)} \sum_{w \in V_q^+} \boldsymbol{h}_{w|u,q}^{(t)} + \mathbf{W}_2^{(t)} \sum_{w \in V_q^-} \boldsymbol{h}_{w|u,q}^{(t)}.$$

We report all results in Table 2. First, we observe that the addition of global readout does not lead to significant differences on WN18RR, but it degrades the model performance on FB15k-237 in two splits (v2 and v4). We attribute this to having more relation types in FB15k-237, so a readout over all nodes does not necessarily inform us about a relation-specific prediction. This is the motivation behind the relation-specific readout: we observe that it leads to significant improvements compared to using no readout (or a global readout) over all splits, leading to state-of-the art results. In contrast, relation-specific readout does not yield improvements on the relationally sparse WN18RR. These findings suggest that the global information can be very useful in practice, assuming the graph is rich in relation types. We further conduct a synthetic experiment to validate the expressive power of using global readout: architectures without global readout cannot exceed random performance, while those using a global readout can solve the synthetic link prediction task (see Appendix E).

## 7 Outlook and limitations

We studied a broad general class of GNNs designed for link prediction over knowledge graphs with a focus on formally characterizing their expressive power. Our study shows that recent state-of-the-art models are at a 'sweet spot': while they compute binary invariants, they do so through some local variations of higher-dimensional WL tests. This is precisely characterized within our framework. To capture global properties, we propose readout components and show that their relational variant is useful on datasets with many relation types. This study paints a more complete picture for our overall understanding of existing models and informs future work on possible directions.

In terms of limitations, it is important to note that the presented approach has certain restrictions. For instance, it is limited to binary tasks, such as link prediction. Thus, this approach does not directly extend to higher arity tasks, e.g., link prediction on relational hypergraphs or predicting the existence of substructures of size $k$ in a knowledge graph. When considering local or conditional variations of the $k$-WL algorithm, computational demands can become a significant challenge. Further research is needed to explore potential optimizations for higher-order MPNNs that can enhance their performance while preserving their expressive power.

## 8 Acknowledgement

The authors would like to acknowledge the use of the University of Oxford Advanced Research Computing (ARC) facility in carrying out this work (http://dx.doi.org/10.5281/zenodo.22558). We would also like to thank Google Cloud for kindly providing computational resources. Barceló is funded by ANID–Millennium Science Initiative Program - CodeICN17002. Romero is funded by Fondecyt grant 11200956. Barceló and Romero are funded by the National Center for Artificial Intelligence CENIA FB210017, BasalANID.

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

# A  Proofs of technical statements

## A.1  Expressive power of R-MPNNs

The expressive power of R-MPNNs has been recently characterized in terms of a relational variant of the Weisfeiler–Leman test [4]. We define the *relational local* $1$-*WL test*[2], which we denote by $\mathsf{rwl}_1$. Let $G = (V, E, R, c)$ be a knowledge graph. For each $t \geq 0$, the test updates the coloring as follows:

$$\mathsf{rwl}_1^{(0)}(v) = c(v),$$
$$\mathsf{rwl}_1^{(t+1)}(v) = \tau\big(\mathsf{rwl}_1^{(t)}(v), \{\!\!\{(\mathsf{rwl}_1^{(t)}(v), r)\,|\, w \in \mathcal{N}_r(v), r \in R\}\!\!\}\big),$$

where $\tau$ injectively maps the above pair to a unique color, which has not been used in previous iterations. Hence, $\mathsf{rwl}_1^{(t)}$ defines a node invariant for all $t \geq 0$.

The following result generalizes results from Barceló et al. [4].

**Theorem A.1.** *Let $G = (V, E, R, c)$ be a knowledge graph.*

1. *For all initial feature maps $\boldsymbol{x}$ with $c \equiv \boldsymbol{x}$, all R-MPNNs with $T$ layers, and $0 \leq t \leq T$, it holds that $\mathsf{rwl}_1^{(t)} \preceq \boldsymbol{h}^{(t)}$.*

2. *For all $T \geq 0$ and history function $f$, there is an initial feature map $\boldsymbol{x}$ with $c \equiv \boldsymbol{x}$ and an R-MPNN without global readout with $T$ layers and history function $f$, such that for all $0 \leq t \leq T$ we have $\mathsf{rwl}_1^{(t)} \equiv \boldsymbol{h}^{(t)}$.*

Intuitively, item (1) states that the relational local $1$-WL algorithm upper bounds the power of any R-MPNN $\mathcal{A}$: if the test cannot distinguish two nodes, then $\mathcal{A}$ cannot either. On the other hand, item (2) states that R-MPNNs can be as expressive as $\mathsf{rwl}_1$: for any time limit $T$, there is an R-MPNN that simulates $T$ iterations of the test. This holds even without a global readout component.

*Remark* A.2. A direct corollary of Theorem A.1 is that R-MPNNs have the same expressive power as $\mathsf{rwl}_1$, *independently* of their history function.

In order to prove Theorem A.1, we define a variant of $\mathsf{rwl}_1$ as follows. For a history function $f : \mathbb{N} \to \mathbb{N}$ (recall $f$ is non-decreasing and $f(t) \leq t$), and given a knowledge graph $G = (V, E, R, c)$, we define the $\mathsf{rwl}_{1,f}$ test via the following update rules:

$$\mathsf{rwl}_{1,f}^{(0)}(v) = c(v)$$
$$\mathsf{rwl}_{1,f}^{(t+1)}(v) = \tau\big(\mathsf{rwl}_{1,f}^{(f(t))}(v), \{\!\!\{(\mathsf{rwl}_{1,f}^{(t)}(v), r)\,|\, w \in \mathcal{N}_r(v), r \in R\}\!\!\}\big),$$

where $\tau$ injectively maps the above pair to a unique color, which has not been used in previous iterations. Note that $\mathsf{rwl}_1$ corresponds to $\mathsf{rwl}_{1,id}$ for the identity function $id(t) = t$.

We have that $\mathsf{rwl}_{1,f}^{(t)}$ is always a refinement of $\mathsf{rwl}_{1,f}^{(t-1)}$. Note that this is trivial for $f(t) = t$ but not for arbitrary history functions.

**Proposition A.3.** *Let $G = (V, E, R, c)$ be a knowledge graph and $f$ be a history function. Then for all $t \geq 0$, we have $\mathsf{rwl}_{1,f}^{(t+1)} \preceq \mathsf{rwl}_{1,f}^{(t)}$.*

*Proof.* We proceed by induction on $t$. For $t = 0$, note that $\mathsf{rwl}_{1,f}^{(1)}(u) = \mathsf{rwl}_{1,f}^{(1)}(v)$ implies $\mathsf{rwl}_{1,f}^{(f(0))}(u) = \mathsf{rwl}_{1,f}^{(f(0))}(v)$ and $f(0) = 0$. For the inductive case, suppose $\mathsf{rwl}_{1,f}^{(t+1)}(u) = \mathsf{rwl}_{1,f}^{(t+1)}(v)$, for $t \geq 1$ and $u, v \in V$. By injectivity of $\tau$, we have that:

$$\mathsf{rwl}_{1,f}^{(f(t))}(u) = \mathsf{rwl}_{1,f}^{(f(t))}(v)$$
$$\{\!\!\{(\mathsf{rwl}_{1,f}^{(t)}(w), r) \mid w \in \mathcal{N}_r(u), r \in R\}\!\!\} = \{\!\!\{(\mathsf{rwl}_{1,f}^{(t)}(w), r) \mid w \in \mathcal{N}_r(v), r \in R\}\!\!\}.$$

---

[2]This test over single-relation graphs is often called *color refinement* [13]. In Barceló et al. [4] it is also called *multi-relational* $1$-*WL*.

By inductive hypothesis and the facts that $f(t-1) \leq f(t)$ (as $f$ is non-decreasing) and $\mathsf{rwl}_{1,f}^{(f(t))}(u) = \mathsf{rwl}_{1,f}^{(f(t))}(v)$, we obtain $\mathsf{rwl}_{1,f}^{(f(t-1))}(u) = \mathsf{rwl}_{1,f}^{(f(t-1))}(v)$. On the other hand, by inductive hypothesis, we obtain that

$$\{\!\!\{(\mathsf{rwl}_{1,f}^{(t-1)}(w), r) \mid w \in \mathcal{N}_r(u), r \in R\}\!\!\} = \{\!\!\{(\mathsf{rwl}_{1,f}^{(t-1)}(w), r) \mid w \in \mathcal{N}_r(v), r \in R\}\!\!\}.$$

We conclude that $\mathsf{rwl}_{1,f}^{(t)}(u) = \mathsf{rwl}_{1,f}^{(t)}(v)$. $\qquad\square$

As it turns out, $\mathsf{rwl}_{1,f}^{(t)}$ defines the same coloring independently of $f$:

**Proposition A.4.** *Let $G = (V, E, R, c)$ be a knowledge graph and $f, f'$ be history functions. Then for all $t \geq 0$, we have that $\mathsf{rwl}_{1,f}^{(t)} \equiv \mathsf{rwl}_{1,f'}^{(t)}$.*

*Proof.* We apply induction on $t$. For $t = 0$, we have $\mathsf{rwl}_{1,f}^{(0)} \equiv c \equiv \mathsf{rwl}_{1,f'}^{(0)}$. For the inductive case, suppose $\mathsf{rwl}_{1,f}^{(t)}(u) = \mathsf{rwl}_{1,f}^{(t)}(v)$, for $t \geq 1$ and $u, v \in V$. Since $f'(t-1) \leq t-1$ and by Proposition A.3, we have that $\mathsf{rwl}_{1,f}^{(f'(t-1))}(u) = \mathsf{rwl}_{1,f}^{(f'(t-1))}(v)$. The inductive hypothesis implies that $\mathsf{rwl}_{1,f'}^{(f'(t-1))}(u) = \mathsf{rwl}_{1,f'}^{(f'(t-1))}(v)$. On the other hand, by injectivity of $\tau$ we have

$$\{\!\!\{(\mathsf{rwl}_{1,f}^{(t-1)}(w), r) \mid w \in \mathcal{N}_r(u), r \in R\}\!\!\} = \{\!\!\{(\mathsf{rwl}_{1,f}^{(t-1)}(w), r) \mid w \in \mathcal{N}_r(v), r \in R\}\!\!\}.$$

The inductive hypothesis implies that

$$\{\!\!\{(\mathsf{rwl}_{1,f'}^{(t-1)}(w), r) \mid w \in \mathcal{N}_r(u), r \in R\}\!\!\} = \{\!\!\{(\mathsf{rwl}_{1,f'}^{(t-1)}(w), r) \mid w \in \mathcal{N}_r(v), r \in R\}\!\!\}.$$

Summing up, we have that $\mathsf{rwl}_{1,f'}^{(t)}(u) = \mathsf{rwl}_{1,f'}^{(t)}(v)$, and hence $\mathsf{rwl}_{1,f}^{(t)} \preceq \mathsf{rwl}_{1,f'}^{(t)}$. The case $\mathsf{rwl}_{1,f'}^{(t)} \preceq \mathsf{rwl}_{1,f}^{(t)}$ follows by symmetry. $\qquad\square$

Now we are ready to prove Theorem A.1.

We start with item (1). Take an initial feature map $\boldsymbol{x}$ with $c \equiv \boldsymbol{x}$, an R-MPNN with $T$ layers, and history function $f$. It suffices to show that $\mathsf{rwl}_{1,f}^{(t)} \preceq \boldsymbol{h}^{(t)}$, for all $0 \leq t \leq T$. Indeed, by Proposition A.4, we have $\mathsf{rwl}_{1,f}^{(t)} \equiv \mathsf{rwl}_{1,id}^{(t)} \equiv \mathsf{rwl}_{1}^{(t)}$, where $id$ is the identity function $id(t) = t$, and hence the result follows. We apply induction on $t$. The case $t = 0$ follows directly as $\mathsf{rwl}_{1,f}^{(0)} \equiv c \equiv \boldsymbol{x} \equiv \boldsymbol{h}^{(0)}$. For the inductive case, assume $\mathsf{rwl}_{1,f}^{(t)}(u) = \mathsf{rwl}_{1,f}^{(t)}(v)$ for $t \geq 1$ and $u, v \in V$. By injectivity of $\tau$ we have:

$$\mathsf{rwl}_{1,f}^{(f(t-1))}(u) = \mathsf{rwl}_{1,f}^{(f(t-1))}(v)$$

$$\{\!\!\{(\mathsf{rwl}_{1,f}^{(t-1)}(w), r) \mid w \in \mathcal{N}_r(u), r \in R\}\!\!\} = \{\!\!\{(\mathsf{rwl}_{1,f}^{(t-1)}(w), r) \mid w \in \mathcal{N}_r(v), r \in R\}\!\!\}.$$

By inductive hypothesis we have (recall $f(t-1) \leq t-1$):

$$\boldsymbol{h}_u^{(f(t-1))} = \boldsymbol{h}_v^{(f(t-1))}$$

$$\{\!\!\{\boldsymbol{h}_w^{(t-1)} \mid w \in \mathcal{N}_r(u)\}\!\!\} = \{\!\!\{\boldsymbol{h}_w^{(t-1)} \mid w \in \mathcal{N}_r(v)\}\!\!\} \qquad \text{for each } r \in R.$$

This implies that $\{\!\!\{\textsc{Msg}_r(\boldsymbol{h}_w^{(t-1)}) \mid w \in \mathcal{N}_r(u)\}\!\!\} = \{\!\!\{\textsc{Msg}_r(\boldsymbol{h}_w^{(t-1)}) \mid w \in \mathcal{N}_r(v)\}\!\!\}$, for each $r \in R$, and hence:

$$\{\!\!\{\textsc{Msg}_r(\boldsymbol{h}_w^{(t-1)}) \mid w \in \mathcal{N}_r(u), r \in R\}\!\!\} = \{\!\!\{\textsc{Msg}_r(\boldsymbol{h}_w^{(t-1)}) \mid w \in \mathcal{N}_r(v), r \in R\}\!\!\}.$$

We conclude that

$$\begin{aligned}
\boldsymbol{h}_u^{(t)} &= \textsc{Upd}\big(\boldsymbol{h}_u^{(f(t-1))}, \textsc{Agg}(\{\!\!\{\textsc{Msg}_r(\boldsymbol{h}_w^{(t-1)}) \mid w \in \mathcal{N}_r(u), r \in R\}\!\!\}), \textsc{Read}(\{\!\!\{\boldsymbol{h}_w^{(t)} \mid w \in V\}\!\!\})\big) \\
&= \textsc{Upd}\big(\boldsymbol{h}_v^{(f(t-1))}, \textsc{Agg}(\{\!\!\{\textsc{Msg}_r(\boldsymbol{h}_w^{(t-1)}) \mid w \in \mathcal{N}_r(v), r \in R\}\!\!\}), \textsc{Read}(\{\!\!\{\boldsymbol{h}_w^{(t)} \mid w \in V\}\!\!\})\big) \\
&= \boldsymbol{h}_v^{(t)}.
\end{aligned}$$

For item (2), we refine the proof of Theorem 2 from Barceló et al. [4], which is based on ideas from Morris et al. [21]. In comparison with Barceló et al. [4], in our case, we have arbitrary adjacency matrices for each relation type, not only symmetric ones, and arbitrary history functions, not only the identity. However, the arguments still apply. Moreover, the most important difference is that here we aim for a model of R-MPNN without global readout that uses a single parameter matrix, instead of two parameter matrices as in Barceló et al. [4] (one for self-representations and the other for neighbors representations). This makes the simulation of $\mathrm{rwl}_1$ more challenging.

We use models of R-MPNNs without global readout of the following form:

$$\boldsymbol{h}_v^{(t+1)} = \mathrm{sign}\left(\boldsymbol{W}^{(t)}\big(\boldsymbol{h}_v^{(f(t))} + \sum_{r \in R}\sum_{w \in \mathcal{N}_r(v)} \alpha_r \boldsymbol{h}_w^{(t)}\big) - \boldsymbol{b}\right),$$

where $\boldsymbol{W}^{(t)}$ is a parameter matrix and $\boldsymbol{b}$ is the bias term (we shall use the all-ones vector $\boldsymbol{b} = \boldsymbol{1}$). As message function $\mathrm{MSG}_r$ we use *vector scaling*, that is, $\mathrm{MSG}_r(\boldsymbol{h}) = \alpha_r \boldsymbol{h}$, where $\alpha_r$ is a parameter of the model. For the non-linearity, we use the sign function $\mathrm{sign}$. We note that the proof also works for the ReLU function, following arguments from Corollary 16 in Morris et al. [21].

For a matrix $\boldsymbol{B}$, we denote by $\boldsymbol{B}_i$ its $i$-th column. Let $n = |V|$ and without loss of generality assume $V = \{1, \ldots, n\}$. We will write features maps $\boldsymbol{h} : V \to \mathbb{R}^d$ for $G = (V, E, R, c)$ also as matrices $\boldsymbol{H} \in \mathbb{R}^{d \times n}$, where the column $\boldsymbol{H}_v$ corresponds to the $d$-dimensional feature vector for $v$. Then we can also write our R-MPNN model in matrix form:

$$\boldsymbol{H}^{(t+1)} = \mathrm{sign}\left(\boldsymbol{W}^{(t)}\big(\boldsymbol{H}^{(f(t))} + \sum_{r \in R} \alpha_r \boldsymbol{H}^{(t)} \boldsymbol{A}_r\big) - \boldsymbol{J}\right),$$

where $\boldsymbol{A}_r$ is the adjacency matrix of $G$ for relation type $r \in R$ and $\boldsymbol{J}$ is the all-ones matrix of appropriate dimensions.

Let $\boldsymbol{Fts}$ be the following $n \times n$ matrix:

$$\boldsymbol{Fts} = \begin{bmatrix} -1 & -1 & \cdots & -1 & -1 \\ 1 & -1 & \ddots & & -1 \\ \vdots & \ddots & \ddots & \ddots & \vdots \\ 1 & & \ddots & -1 & -1 \\ 1 & 1 & \cdots & 1 & -1 \end{bmatrix}$$

That is, $(\boldsymbol{Fts})_{ij} = -1$ if $j \geq i$, and $(\boldsymbol{Fts})_{ij} = 1$ otherwise. Note that the columns of $\boldsymbol{Fts}$ are linearly independent. We shall use the columns of $\boldsymbol{Fts}$ as node features in our simulation.

The following lemma is an adaptation of Lemma 9 from Morris et al. [21].

**Lemma A.5.** *Let $\boldsymbol{B} \in \mathbb{N}^{n \times p}$ be a matrix such that $p \leq n$, and all the columns are pairwise distinct and different from the all-zeros column. Then there is a matrix $\boldsymbol{X} \in \mathbb{R}^{n \times n}$ such that the matrix $\mathrm{sign}(\boldsymbol{XB} - \boldsymbol{J}) \in \{-1, 1\}^{n \times p}$ is precisely the sub-matrix of $\boldsymbol{Fts}$ given by its first $p$ columns.*

*Proof.* Let $\boldsymbol{z} = (1, m, m^2, \ldots, m^{n-1}) \in \mathbb{N}^{1 \times n}$, where $m$ is the largest entry in $\boldsymbol{B}$, and $\boldsymbol{b} = \boldsymbol{zB} \in \mathbb{N}^{1 \times p}$. By construction, the entries of $\boldsymbol{b}$ are positive and pairwise distinct. Without loss of generality, we assume that $\boldsymbol{b} = (b_1, b_2, \ldots, b_p)$ for $b_1 > b_2 > \cdots > b_p > 0$. As the $b_i$ are ordered, we can choose numbers $x_1, \ldots, x_p \in \mathbb{R}$ such that $b_i \cdot x_j < 1$ if $i \geq j$, and $b_i \cdot x_j > 1$ if $i < j$, for all $i, j \in \{1, \ldots, p\}$. Let $\boldsymbol{x} = (x_1, \ldots, x_p, 2/b_p, \ldots, 2/b_p)^T \in \mathbb{R}^{n \times 1}$. Note that $(2/b_p) \cdot b_i > 1$, for all $i \in \{1, \ldots, p\}$. Then $\mathrm{sign}(\boldsymbol{xb} - \boldsymbol{J})$ is precisely the sub-matrix of $\boldsymbol{Fts}$ given by its first $p$ columns. We can choose $\boldsymbol{X} = \boldsymbol{xz} \in \mathbb{R}^{n \times n}$. $\qquad\square$

Now we are ready to show item (2). Let $f$ be any history function and $T \geq 0$. It suffices to show that there is a feature map $\boldsymbol{x}$ with $c \equiv \boldsymbol{x}$ and an R-MPNN without global readout with $T$ layers and history function $f$ such that $\mathrm{rwl}_{1,f}^{(t)} \equiv \boldsymbol{h}^{(t)}$, for all $0 \leq t \leq T$. Indeed, by Proposition A.4, we have $\mathrm{rwl}_{1,f}^{(t)} \equiv \mathrm{rwl}_{1,id}^{(t)} \equiv \mathrm{rwl}_1^{(t)}$, where $id$ is the identity function $id(t) = t$, and then the result follows. We conclude item (2) by showing the following lemma:

**Lemma A.6.** *There is a feature map $h^{(0)} : V \to \mathbb{R}^n$, and for all $0 \leq t < T$, there is a feature map $h^{(t+1)} : V \to \mathbb{R}^n$, a matrix $W^{(t)} \in \mathbb{R}^{n \times n}$ and scaling factors $\alpha_r^{(t)} \in \mathbb{R}$, for each $r \in R$, such that:*

- $h^{(t)} \equiv \mathsf{rwl}_{1,f}^{(t)}$.

- *The columns of $H^{(t)} \in \mathbb{R}^{n \times n}$ are columns of $Fts$ (recall $H^{(t)}$ is the matrix representation of $h^{(t)}$).*

- $H^{(t+1)} = \mathrm{sign}\left( W^{(t)} \big( H^{(f(t))} + \sum_{r \in R} \alpha_r^{(t)} H^{(t)} A_r \big) - J \right)$.

*Proof.* We proceed by induction on $t$. Suppose that the node coloring $\mathsf{rwl}_{1,f}^{(0)} \equiv c$ uses colors $1, \ldots, p$, for $p \leq n$. Then we choose $h^{(0)}$ (this is the initial feature map $x$ in the statement of item (2)) such that $h_v^{(0)} = Fts_{c(v)}$, that is, $h_v^{(0)}$ is the $c(v)$-th column of $Fts$. We have that $h^{(0)}$ satisfies the required conditions.

For the inductive case, assume that $h^{(t')} \equiv \mathsf{rwl}_{1,f}^{(t')}$ and the columns of $H^{(t')}$ are columns of $Fts$, for all $0 \leq t' \leq t < T$. We need to find $h^{(t+1)}$, $W^{(t)}$ and $\alpha_r^{(t)}$ satisfying the conditions. Let $M \in \mathbb{R}^{n \times n}$ be the matrix inverse of $Fts$. If $H_v^{(t')}$ is the $i$-th column of $Fts$, we say that $v$ has color $i$ at iteration $t'$. Observe that for all $0 \leq t' \leq t$, we have

$$(MH^{(t')})_{iv} = \begin{cases} 1 & \text{if } v \text{ has color } i \text{ at iteration } t' \\ 0 & \text{otherwise.} \end{cases}$$

In other words, the $v$-th column of $MH^{(t')}$ is simply a one-hot encoding of the color of $v$ at iteration $t'$. For each $r \in R$ we have

$$(MH^{(t)} A_r)_{iv} = |\{w \in \mathcal{N}_r(v) \mid w \text{ has color } i \text{ at iteration } t\}|.$$

Hence the $v$-th column of $MH^{(t)} A_r$ is an encoding of the multiset of colors for the neighborhood $\mathcal{N}_r(v)$, at iteration $t$. Let $r_1, \ldots, r_m$ be an enumeration of the relation types in $R$. Let $D \in \mathbb{R}^{(m+1)n \times n}$ be the matrix obtained by horizontally concatenating the matrices $MH^{(f(t))}$, $MH^{(t)} A_{r_1}, \ldots, MH^{(t)} A_{r_m}$. Since $H^{(f(t))} \equiv \mathsf{rwl}_{1,f}^{(f(t))}$ and $H^{(t)} \equiv \mathsf{rwl}_{1,f}^{(t)}$, we have that $D \equiv \mathsf{rwl}_{1,f}^{(t+1)}$. Now note that $D \equiv E$, where

$$E = MH^{(f(t))} + \sum_{i=1}^{m} (n+1)^i MH^{(t)} A_{r_i}.$$

Indeed, $E_{iv}$ is simply the $(n + 1)$-base representation of the vector $(D_{iv}, D_{(n+i)v}, D_{(2n+i)v} \ldots, D_{(mn+i)v})$, and hence $E_u = E_v$ if and only if $D_u = D_v$ (note that the entries of $D$ are in $\{0, \ldots, n\}$). In particular, $E \equiv \mathsf{rwl}_{1,f}^{(t+1)}$.

Let $p$ be the number of distinct columns of $E$ and let $\widetilde{E} \in \mathbb{N}^{n \times p}$ be the matrix whose columns are the distinct columns of $E$ in an arbitrary but fixed order. We can apply Lemma A.5 to $\widetilde{E}$ and obtain a matrix $X \in \mathbb{R}^{n \times n}$ such that $\mathrm{sign}(X\widetilde{E} - J)$ is precisely the sub-matrix of $Fts$ given by its first $p$ columns. We choose $H^{(t+1)} = \mathrm{sign}(XE - J) \in \mathbb{R}^{n \times n}$, $W^{(t)} = XM \in \mathbb{R}^{n \times n}$ and $\alpha_{r_i}^{(t)} = (n+1)^i$. Note that the columns of $H^{(t+1)}$ are columns of $Fts$, and that $H^{(t+1)} \equiv E \equiv \mathsf{rwl}_{1,f}^{(t+1)}$. Finally, we have

$$\begin{aligned} H^{(t+1)} &= \mathrm{sign}(XE - J) \\ &= \mathrm{sign}(X\big(MH^{(f(t))} + \sum_{i=1}^{m} (n+1)^i MH^{(t)} A_{r_i}\big) - J) \\ &= \mathrm{sign}(W^{(t)}\big(H^{(f(t))} + \sum_{i=1}^{m} \alpha_{r_i}^{(t)} H^{(t)} A_{r_i}\big) - J). \end{aligned}$$

$\square$

Note that our result applies to more complex message functions such as $\text{MSG}_r(\boldsymbol{h}) = \boldsymbol{h} * \boldsymbol{b}_r$, where $*$ stands for element-wise multiplication and $\boldsymbol{b}_r$ is a vector parameter, and $\text{MSG}_r(\boldsymbol{h}) = \boldsymbol{W}_r \boldsymbol{h}$, where $\boldsymbol{W}_r$ is matrix parameter, as they can easily express vector scaling. The first case has been used for the model CompGCN [32], while the second case has been used for R-GCN [28].

## A.2 Proof of Theorem 5.1

We recall the statement of the theorem:

**Theorem 5.1.** *Let $G = (V, E, R, \boldsymbol{x}, \eta)$ be a knowledge graph, where $\boldsymbol{x}$ is a feature map, and $\eta$ is a pairwise coloring satisfying target node distinguishability. Let $q \in R$ be any query relation. Then:*

1. *For all C-MPNNs with $T$ layers and initialization $\text{INIT}$ satisfying $\text{INIT} \equiv \eta$, and all $0 \le t \le T$, we have $\text{rawl}_2^{(t)} \preceq \boldsymbol{h}_q^{(t)}$.*

2. *For all $T \ge 0$ and history function $f$, there is an C-MPNN without global readout with $T$ layers and history function $f$ such that for all $0 \le t \le T$, we have $\text{rawl}_2^{(t)} \equiv \boldsymbol{h}_q^{(t)}$.*

We start by showing item (1) and (2) of the theorem for the case without global readout. As we explain later, extending item (1) to include readouts is straightforward. In order to prove this particular case, we apply a reduction to R-MPNNs and the relational local 1-WL. Before doing so we need some auxiliary results.

Let $G = (V, E, R, c, \eta)$ be a knowledge graph where $\eta$ is a pairwise coloring. We denote by $G^2$ the knowledge graph $G^2 = (V \times V, E', R, c_\eta)$ where $E' = \{r((u, w), (u, v)) \mid r(w, v) \in E, r \in R\}$ and $c_\eta$ is the node coloring $c_\eta((u, v)) = \eta(u, v)$. Note that the coloring $c$ is irrelevant in the construction. Intuitively, $G^2$ encodes the adjacency relation between pairs of nodes of $G$ used in C-MPNNs. We stress that $G$ and $G^2$ have the same relation type set $R$. If $\mathcal{A}$ is a C-MPNN and $\mathcal{B}$ is an R-MPNN, we write $\boldsymbol{h}_{\mathcal{A},G}^{(t)}(u, v) := \boldsymbol{h}_q^{(t)}(u, v)$ and $\boldsymbol{h}_{\mathcal{B},G^2}^{(t)}((u, v)) := \boldsymbol{h}^{(t)}((u, v))$ for the features computed by $\mathcal{A}$ and $\mathcal{B}$ over $G$ and $G^2$, respectively. We sometimes write $\mathcal{N}_r^H(v)$ to emphasize that the neighborhood is taken over the knowledge graph $H$. Finally, we say that an initial feature map $\boldsymbol{y}$ for $G^2$ satisfies target node distinguishability if $\boldsymbol{y}((u, u)) \ne \boldsymbol{y}((u, v))$ for all $u \ne v$.

We have the following equivalence between C-MPNNs and R-MPNNs without global readout:

**Proposition A.7.** *Let $G = (V, E, R, \boldsymbol{x}, \eta)^3$ be a knowledge graph where $\boldsymbol{x}$ is a feature map, and $\eta$ is a pairwise coloring. Let $q \in R$ be any query relation. Then:*

1. *For every C-MPNN without global readout $\mathcal{A}$ with $T$ layers, there is an initial feature map $\boldsymbol{y}$ for $G^2$ an R-MPNN without global readout $\mathcal{B}$ with $T$ layers such that for all $0 \le t \le T$ and $u, v \in V$, we have $\boldsymbol{h}_{\mathcal{A},G}^{(t)}(u, v) = \boldsymbol{h}_{\mathcal{B},G^2}^{(t)}((u, v))$.*

2. *For every initial feature map $\boldsymbol{y}$ for $G^2$ satisfying target node distinguishability and every R-MPNN without global readout $\mathcal{B}$ with $T$ layers, there is a C-MPNN without global readout $\mathcal{A}$ with $T$ layers such that for all $0 \le t \le T$ and $u, v \in V$, we have $\boldsymbol{h}_{\mathcal{A},G}^{(t)}(u, v) = \boldsymbol{h}_{\mathcal{B},G^2}^{(t)}((u, v))$.*

*Proof.* We start with item (1). The sought R-MPNN $\mathcal{B}$ has the same history function and the same message, aggregation, and update functions as $\mathcal{A}$ for all the $T$ layers. The initial feature map $\boldsymbol{y}$ is defined as $\boldsymbol{y}((u, v)) = \text{INIT}(u, v, q)$, where $\text{INIT}$ is the initialization function of $\mathcal{A}$.

We show the equivalence by induction on $t$. For $t = 0$, we have $\boldsymbol{h}_{\mathcal{A}}^{(0)}(u, v) = \text{INIT}(u, v, q) = \boldsymbol{y}((u, v)) = \boldsymbol{h}_{\mathcal{B}}^{(0)}((u, v))$. For the inductive case, take $u, v \in V$. We have

$$\boldsymbol{h}_{\mathcal{A}}^{(t+1)}(u, v) = \text{UPD}\big(\boldsymbol{h}_{\mathcal{A}}^{(f(t))}(u, v), \text{AGG}(\{\!\!\{\text{MSG}_r(\boldsymbol{h}_{\mathcal{A}}^{(t)}(u, w), \boldsymbol{z}_q) \mid w \in \mathcal{N}_r^G(v), r \in R\}\!\!\})\big)$$
$$= \text{UPD}\big(\boldsymbol{h}_{\mathcal{B}}^{(f(t))}((u, v)), \text{AGG}(\{\!\!\{\text{MSG}_r(\boldsymbol{h}_{\mathcal{B}}^{(t)}((u, w)), \boldsymbol{z}_q) \mid (u, w) \in \mathcal{N}_r^{G^2}((u, v)), r \in R\}\!\!\})\big)$$
$$= \boldsymbol{h}_{\mathcal{B}}^{(t+1)}((u, v)).$$

For item (2), we take $\mathcal{A}$ to have the same history function and the same message, aggregation, and update functions than $\mathcal{B}$, for all the $T$ layers, and initialization function $\text{INIT}$ such that $\text{INIT}(u, v, q) = \boldsymbol{y}((u, v))$. The argument for the equivalence is the same as item (1). $\qquad\square$

---

[3] The pairwise coloring $\eta$ does not play any role in the proposition.

Regarding WL algorithms, we have a similar equivalence:

**Proposition A.8.** *Let $G = (V, E, R, c, \eta)$ be a knowledge graph where $\eta$ is a pairwise coloring. For all $t \geq 0$ and $u, v \in V$, we have that $\mathsf{rawl}_2^{(t)}(u, v)$ computed over $G$ coincides with $\mathsf{rwl}_1^{(t)}((u, v))$ computed over $G^2 = (V \times V, E', R, c_\eta)$.*

*Proof.* For $t = 0$, we have $\mathsf{rawl}_2^{(0)}(u, v) = \eta(u, v) = c_\eta((u, v)) = \mathsf{rwl}_1^{(0)}((u, v))$. For the inductive case, we have

$$\mathsf{rawl}_2^{(t+1)}(u, v) = \tau\big(\mathsf{rawl}_2^{(t)}(u, v), \{\!\{(\mathsf{rawl}_2^{(t)}(u, w), r) \mid w \in \mathcal{N}_r^G(v), r \in R)\}\!\}\big)$$
$$= \tau\big(\mathsf{rwl}_1^{(t)}((u, v)), \{\!\{(\mathsf{rwl}_1^{(t)}((u, w)), r) \mid (u, w) \in \mathcal{N}_r^{G^2}((u, v)), r \in R)\}\!\}\big)$$
$$= \mathsf{rwl}_1^{(t+1)}((u, v)).$$

$\square$

Now we are ready to prove Theorem 5.1.

For $G = (V, E, R, \boldsymbol{x}, \eta)$, we consider $G^2 = (V \times V, E', R, c_\eta)$. We start with item (1). Let $\mathcal{A}$ be a C-MPNN without global readout with $T$ layers and initialization INIT satisfying INIT $\equiv \eta$ and let $0 \leq t \leq T$. Let $\boldsymbol{y}$ be an initial feature map for $G^2$ and $\mathcal{B}$ be an R-MPNN without global readout with $T$ layers as in Proposition A.7, item (1). Note that $\boldsymbol{y} \equiv c_\eta$ as $\boldsymbol{y}((u, v)) = \text{INIT}(u, v, q)$. We can apply Theorem A.1, item (1) to $G^2$, $\boldsymbol{y}$ and $\mathcal{B}$ and conclude that $\mathsf{rwl}_1^{(t)} \preceq \boldsymbol{h}_{\mathcal{B}, G^2}^{(t)}$. This implies that $\mathsf{rawl}_1^{(t)} \preceq \boldsymbol{h}_{\mathcal{A}, G}^{(t)}$.

For item (2), let $T \geq 0$ and $f$ be a history function. We can apply Theorem A.1, item (2), to $G^2$ to obtain an initial feature map $\boldsymbol{y}$ with $\boldsymbol{y} \equiv c_\eta$ and an R-MPNN without global readout $\mathcal{B}$ with $T$ layers and history function $f$ such that for all $0 \leq t \leq T$, we have $\mathsf{rwl}_1^{(t)} \equiv \boldsymbol{h}_{\mathcal{B}, G^2}^{(t)}$. Note that $\boldsymbol{y}$ satisfies target node distinguishability since $\eta$ does. Let $\mathcal{A}$ be the C-MPNN without global readout obtained from Proposition A.7, item (2). We have that $\mathsf{rawl}_2^{(t)} \equiv \boldsymbol{h}_{\mathcal{A}, G}^{(t)}$ as required.

*Remark* A.9. Note that item (2) holds for the *basic* model of C-MPNNs (without global readout), presented in Section 4.1, with the three proposed message functions $\text{MSG}_r^1, \text{MSG}_r^2, \text{MSG}_r^3$, as they can express vector scaling.

Finally, note that the extension of item (1) in Theorem 5.1 to the case with global readouts is straightforward. Indeed, we can apply exactly the same argument we used for the case without global readout. As for two pairs of the form $(u, v)$ and $(u, v')$ the global readout vectors are identical (to $\text{READ}(\{\!\{\boldsymbol{h}_{w|u, q}^{(t)} \mid w \in V\}\!\})$), this argument still works.

## A.3 Proof of Theorem 5.2

We recall the statement of the theorem:

**Theorem 5.2.** *A logical binary classifier is captured by C-MPNNs without global readout if and only if it can be expressed in* $\mathsf{rFO}_{\text{cnt}}^3$.

The proof is a reduction to the one-dimensional case, that is, R-MPNNs without global readout and a logic denoted by $\mathsf{rFO}_{\text{cnt}}^2$. So in order to show Theorem 5.2 we need some definitions and auxiliary results.

Fix a set of relation types $R$ and a set of node colors $\mathcal{C}$. We consider knowledge graphs of the form $G = (V, E, R, c)$ where $c$ is a node coloring assigning colors from $\mathcal{C}$. In this context, logical formulas can refer to relation types from $R$ and node colors from $\mathcal{C}$. Following Barceló et al. [3], a *logical (node) classifier* is a unary formula expressible in first-order logic (FO), classifying each node $u$ on a knowledge graph $G$ according to whether the formula holds or not for $u$ over $G$.

We define a fragment $\mathsf{rFO}_{\text{cnt}}^2$ of FO as follows. A $\mathsf{rFO}_{\text{cnt}}^2$ formula is either $a(x)$ for $a \in \mathcal{C}$, or one of the following, where $\varphi$ and $\psi$ are $\mathsf{rFO}_{\text{cnt}}^2$ formulas, $N \geq 1$ is a positive integer and $r \in R$:

$$\neg\varphi(x), \quad \varphi(x) \wedge \psi(x), \quad \exists^{\geq N} y\,(\varphi(y) \wedge r(y, x)).$$

We remark that $\mathsf{rFO}^2_{\mathrm{cnt}}$ is actually the fragment of FO used in Barceló et al. [3] to characterize GNNs, adapted to multiple relations. It is well-known that $\mathsf{rFO}^2_{\mathrm{cnt}}$ is equivalent to *graded modal logic* [8].

The following proposition provides useful translations from $\mathsf{rFO}^2_{\mathrm{cnt}}$ to $\mathsf{rFO}^3_{\mathrm{cnt}}$ and vice versa. Recall from Section A.2, that given a knowledge graph $G = (V, E, R, \eta)$ where $\eta$ is a pairwise coloring, we define the knowledge graph $G^2 = (V \times V, E', R, c_\eta)$ where $E' = \{r((u,w),(u,v)) \mid r(w,v) \in E, r \in R\}$ and $c_\eta$ is the node coloring $c_\eta((u,v)) = \eta(u,v)$.

**Proposition A.10.** *We have the following:*

1. *For all $\mathsf{rFO}^3_{\mathrm{cnt}}$ formula $\varphi(x,y)$, there is a formula $\tilde{\varphi}(x)$ in $\mathsf{rFO}^2_{\mathrm{cnt}}$ such that for all knowledge graph $G = (V, E, R, \eta)$, we have $G, u, v \models \varphi$ if and only if $G^2, (u,v) \models \tilde{\varphi}$.*

2. *For all formula $\varphi(x)$ in $\mathsf{rFO}^2_{\mathrm{cnt}}$, there is a $\mathsf{rFO}^3_{\mathrm{cnt}}$ formula $\tilde{\varphi}(x,y)$ such that for all knowledge graph $G = (V, E, R, \eta)$, we have $G, u, v \models \tilde{\varphi}$ if and only if $G^2, (u,v) \models \varphi$.*

*Proof.* We start with item (1). We define $\tilde{\varphi}(x)$ by induction on the formula $\varphi(x,y)$:

1. If $\varphi(x,y) = a(x,y)$ for color $a$, then $\tilde{\varphi}(x) = a(x)$.

2. If $\varphi(x,y) = \neg\psi(x,y)$, then $\tilde{\varphi}(x) = \neg\tilde{\psi}(x)$.

3. If $\varphi(x,y) = \varphi_1(x,y) \wedge \varphi_2(x,y)$, then $\tilde{\varphi}(x) = \tilde{\varphi}_1(x) \wedge \tilde{\varphi}_2(x)$.

4. If $\varphi(x,y) = \exists^{\geq N} z\, (\psi(x,z) \wedge r(z,y))$ then $\tilde{\varphi}(x) = \exists^{\geq N} y\, (\tilde{\psi}(y) \wedge r(y,x))$.

Fix $G = (V, E, R, \eta)$ and $G^2 = (V \times V, E', R, c_\eta)$. We show by induction on the formula $\varphi$ that $G, u, v \models \varphi$ if and only if $G^2, (u,v) \models \tilde{\varphi}$.

For the base case, that is, case (1) above, we have that $\varphi(x,y) = a(x,y)$ and hence $G, u, v \models \varphi$ iff $\eta(u,v) = a$ iff $c_\eta((u,v)) = a$ iff $G^2, (u,v) \models \tilde{\varphi}$.

Now we consider the inductive case. For case (2) above, we have $\varphi(x,y) = \neg\psi(x,y)$. Then $G, u, v \models \varphi$ iff $G, u, v \not\models \psi$ iff $G^2, (u,v) \not\models \tilde{\psi}$ iff $G^2, (u,v) \models \tilde{\varphi}$.

For case (3), we have $\varphi(x,y) = \varphi_1(x,y) \wedge \varphi_2(x,y)$. Then $G, u, v \models \varphi$ iff $G, u, v \models \varphi_1$ and $G, u, v \models \varphi_2$ iff $G^2, (u,v) \models \tilde{\varphi}_1$ and $G^2, (u,v) \models \tilde{\varphi}_2$ iff $G^2, (u,v) \models \tilde{\varphi}$.

Finally, for case (4), we have $\varphi(x,y) = \exists^{\geq N} z\, (\psi(x,z) \wedge r(z,y))$. Assume $G, u, v \models \varphi$, then there exist at least $N$ nodes $w \in V$ such that $G, u, w \models \psi$ and $r(w,v) \in E$. By the definition of $G^2$, there exist at least $N$ nodes in $G^2$ of the form $(u,w)$ such that $G^2, (u,w) \models \tilde{\psi}$ and $r((u,w),(u,v)) \in E'$. It follows that $G^2, (u,v) \models \tilde{\varphi}$. On the other hand, suppose $G^2, (u,v) \models \tilde{\varphi}$. Then there exist at least $N$ nodes $(o,o')$ in $G^2$ such that $G, (o,o') \models \tilde{\psi}$ and $r((o,o'),(u,v)) \in E'$. By definition of $G^2$ each $(o,o')$ must be of the form $(o,o') = (u,w)$ for some $w \in V$ such that $r(w,v)$. Then there are at least $N$ nodes $w \in V$ such that $G, u, w \models \psi$ and $r(w,v) \in E$. It follows that $G, u, v \models \varphi$.

Item (2) is similar. We define $\tilde{\varphi}(x,y)$ by induction on the formula $\varphi(x)$:

1. If $\varphi(x) = a(x)$ for color $a$, then $\tilde{\varphi}(x,y) = a(x,y)$.

2. If $\varphi(x) = \neg\psi(x)$, then $\tilde{\varphi}(x,y) = \neg\tilde{\psi}(x,y)$.

3. If $\varphi(x) = \varphi_1(x) \wedge \varphi_2(x)$, then $\tilde{\varphi}(x,y) = \tilde{\varphi}_1(x,y) \wedge \tilde{\varphi}_2(x,y)$.

4. If $\varphi(x) = \exists^{\geq N} y\, (\psi(y) \wedge r(y,x))$ then $\tilde{\varphi}(x,y) = \exists^{\geq N} z\, (\tilde{\psi}(x,z) \wedge r(z,y))$.

Following the same inductive argument from item (1), we obtain that $G, u, v \models \tilde{\varphi}$ if and only if $G^2, (u,v) \models \varphi$. $\qquad\square$

The following theorem is an adaptation of Theorem 4.2 from Barceló et al. [3]. The main difference with Barceló et al. [3] is that here we need to handle multiple relation types.

**Theorem A.11.** *A logical classifier is captured by R-MPNNs without global readout if and only if it can be expressed in $\mathsf{rFO}^2_{\mathrm{cnt}}$.*

*Proof.* We start with the backward direction. Let $\varphi(x)$ be a formula in $\mathrm{rFO}^2_{\mathrm{cnt}}$ for relation types $R$ and node colors $\mathcal{C}$. Let $\varphi_1, \ldots, \varphi_L$ be an enumeration of the subformulas of $\varphi$ such that if $\varphi_i$ is a subformula of $\varphi_j$, then $i \leq j$. In particular, $\varphi_L = \varphi$. We shall define an R-MPNN without global readout $\mathcal{B}_\varphi$ with $L$ layers computing $L$-dimensional features in each layer. The idea is that at layer $\ell \in \{1, \ldots, L\}$, the $\ell$-th component of the feature $\boldsymbol{h}_v^{(\ell)}$ is computed correctly and corresponds to 1 if $\varphi_\ell$ is satisfied in node $v$, and 0 otherwise. We add an additional final layer that simply outputs the last component of the feature vector.

We use models of R-MPNNs of the following form:

$$\boldsymbol{h}_v^{(t+1)} = \sigma\Big(\boldsymbol{W}\boldsymbol{h}_v^{(t)} + \sum_{r \in R} \sum_{w \in \mathcal{N}_r(v)} \boldsymbol{W}_r \boldsymbol{h}_w^{(t)} + \boldsymbol{b}\Big),$$

where $\boldsymbol{W} \in \mathbb{R}^{L \times L}$ is a parameter matrix and $\boldsymbol{b} \in \mathbb{R}^L$ is the bias term. As message function $\mathrm{MSG}_r$ we use $\mathrm{MSG}_r(\boldsymbol{h}) = \boldsymbol{W}_r \boldsymbol{h}$, where $\boldsymbol{W}_r \in \mathbb{R}^{L \times L}$ is a parameter matrix . For the non-linearity $\sigma$ we use the truncated ReLU function $\sigma(x) = \min(\max(0, x), 1)$. The $\ell$-th row of $\boldsymbol{W}$ and $\boldsymbol{W}_r$, and the $\ell$-th entry of $\boldsymbol{b}$ are defined as follows (omitted entries are 0):

1. If $\varphi_\ell(x) = a(x)$ for a color $a \in \mathcal{C}$, then $\boldsymbol{W}_{\ell\ell} = 1$.

2. If $\varphi_\ell(x) = \neg\varphi_k(x)$ then $\boldsymbol{W}_{\ell k} = -1$, and $b_\ell = 1$.

3. If $\varphi_\ell(x) = \varphi_j(x) \wedge \varphi_k(x)$ then $\boldsymbol{W}_{\ell j} = 1$, $\boldsymbol{W}_{\ell k} = 1$ and $b_\ell = -1$.

4. If $\varphi_\ell(x) = \exists^{\geq N} y\, (\varphi_k(y) \wedge r(y, x))$ then $(\boldsymbol{W}_r)_{\ell k} = 1$ and $b_\ell = -N + 1$.

Let $G = (V, E, R, c)$ be a knowledge graph with node colors from $\mathcal{C}$. In order to apply $\mathcal{B}_\varphi$ to $G$, we choose initial $L$-dimensional features $\boldsymbol{h}_v^{(0)}$ such that $(\boldsymbol{h}_v^{(0)})_\ell = 1$ if $\varphi_\ell = a(x)$ and $a$ is the color of $v$, and $(\boldsymbol{h}_v^{(0)})_\ell = 0$ otherwise. In other words, the $L$-dimensional initial feature $\boldsymbol{h}_v^{(0)}$ is a one-hot encoding of the color of $v$. It follows from the same arguments than Proposition 4.1 in Barceló et al. [3] that for all $\ell \in \{1, \ldots, L\}$ we have $(\boldsymbol{h}_v^{(t)})_\ell = 1$ if $G, v \models \varphi_\ell$ and $(\boldsymbol{h}_v^{(t)})_\ell = 0$ otherwise, for all $v \in V$ and $t \in \{\ell, \ldots, L\}$. In particular, after $L$ layers, $\mathcal{B}_\varphi$ calculates $\boldsymbol{h}_v^{(L)}$ such that $(\boldsymbol{h}_v^{(L)})_L = 1$ if $G, v \models \varphi$ and $(\boldsymbol{h}_v^{(L)})_L = 0$ otherwise. As layer $L + 1$ extracts the $L$-th component of the feature vector, the result follows.

For the forward direction, we follow the strategy of Theorem 4.2 from Barceló et al. [3]. Given a knowledge graph $G = (V, E, R, c)$ and a number $L \in \mathbb{N}$, we define the *unravelling* of $v \in V$ at depth $L$, denoted $\mathrm{Unr}_G^L(v)$ is the knowledge graph having:

- A node $(v, u_1, \ldots, u_i)$ for each directed path $u_i, \ldots, u_1, v$ in $G$ of length $i \leq L$.

- For each $r \in R$, a fact $r((v, u_1, \ldots, u_i), (v, u_1, \ldots, u_{i-1}))$ for all facts $r(u_i, u_{i-1}) \in E$ (here $u_0 := v$).

- Each node $(v, u_1, \ldots, u_i)$ is colored with $c(u_i)$, that is, the same color as $u_i$.

Note that the notion of directed path is defined in the obvious way, as for directed graphs but ignoring relation types. Note also that $\mathrm{Unr}_G^L(v)$ is a tree in the sense that the underlying undirected graph is a tree.

The following proposition is a trivial adaptation of Observation C.3 from Barceló et al. [3]. We write $\mathrm{Unr}_G^L(v) \simeq \mathrm{Unr}_{G'}^L(v')$ if there exists an isomorphism $f$ from $\mathrm{Unr}_G^L(v)$ to $\mathrm{Unr}_{G'}^L(v')$ such that $f(v) = v'$.

**Proposition A.12.** *Let $G$ and $G'$ be two knowledge graphs and $v$ and $v'$ be nodes in $G$ and $G'$, respectively. Then, for all $L \in \mathbb{N}$, we have that $\mathrm{rwl}_1^{(L)}(v)$ on $G$ coincides with $\mathrm{rwl}_1^{(L)}(v')$ on $G'$ if and only if $\mathrm{Unr}_G^L(v) \simeq \mathrm{Unr}_{G'}^L(v')$.*

As a consequence of Theorem A.1, we obtain:

**Proposition A.13.** *Let $G$ and $G'$ be two knowledge graphs and $v$ and $v'$ be nodes in $G$ and $G'$, respectively, such that $\mathsf{Unr}_G^L(v) \simeq \mathsf{Unr}_{G'}^L(v')$ for all $L \in \mathbb{N}$. Then for any R-MPNN without global readout with $T$ layers, we have that $\boldsymbol{h}_v^{(T)}$ on $G$ coincides with $\boldsymbol{h}_{v'}^{(T)}$ on $G'$.*

Finally, the following theorem follows from Theorem C.5 in Barceló et al. [3], which in turn follows from Theorem 2.2 in Otto [23]. The key observation here is that the results from Otto [23] are actually presented for multi-modal logics, that is, multiple relation types.

**Theorem A.14.** *[23] Let $\alpha$ be a unary FO formula over knowledge graphs. If $\alpha$ is not equivalent to a $\mathsf{rFO}_{\mathrm{cnt}}^2$ formula, then there exist two knowledge graphs $G$ and $G'$, and two nodes $v$ in $G$ and $v'$ in $G'$ such that $\mathsf{Unr}_G^L(v) \simeq \mathsf{Unr}_{G'}^L(v')$ for all $L \in \mathbb{N}$ and such that $G, v \models \alpha$ but $G', v' \not\models \alpha$.*

Now we are ready to obtain the forward direction of the theorem. Suppose that a logical classifier $\alpha$ is captured by an R-MPNN without global readout $\mathcal{B}$ with $T$ layers, and assume by contradiction that $\alpha$ is not equivalent to a $\mathsf{rFO}_{\mathrm{cnt}}^2$ formula. Then we can apply Theorem A.14 and obtain two knowledge graphs $G$ and $G'$, and two nodes $v$ in $G$ and $v'$ in $G'$ such that $\mathsf{Unr}_G^L(v) \simeq \mathsf{Unr}_{G'}^L(v')$ for all $L \in \mathbb{N}$ and such that $G, v \models \alpha$ but $G', v' \not\models \alpha$. Applying Proposition A.13, we have that $\boldsymbol{h}_v^{(T)}$ on $G$ coincides with $\boldsymbol{h}_{v'}^{(T)}$ on $G'$, and hence $\mathcal{B}$ classifies either both $v$ and $v'$ as true over $G$ and $G'$, respectively, or both as false. This is a contradiction. $\qquad\square$

Now Theorem 5.2 follows easily. Let $\alpha$ be a logical binary classifier and suppose it is captured by a C-MPNN without global readout $\mathcal{A}$. By Proposition A.7, item (1), we know that $\mathcal{A}$ can be simulated by a R-MPNN without global readout $\mathcal{B}$ over $G^2$. In turn, we can apply Theorem A.11 and obtain a formula $\varphi$ in $\mathsf{rFO}_{\mathrm{cnt}}^2$ equivalent to $\mathcal{B}$. Finally, we can apply the translation in Proposition A.10, item (2), and obtain a corresponding formula $\tilde{\varphi}$ in $\mathsf{rFO}_{\mathrm{cnt}}^3$. We claim that $\tilde{\varphi}$ captures $\mathcal{A}$. Let $G$ be a knowledge graph and $u, v$ two nodes. We have that $G, u, v \models \tilde{\varphi}$ iff $G^2, (u, v) \models \varphi$ iff $\mathcal{B}$ classifies $(u, v)$ as true over $G^2$ iff $\mathcal{A}$ classifies $(u, v)$ as true over $G$.

The other direction is obtained analogously following the reverse translations.

### A.4  Proof of Theorem 5.3

We recall the statement of the theorem:

**Theorem 5.3.** *Each logical binary classifier expressible in $\mathsf{erFO}_{\mathrm{cnt}}^3$ can be captured by a C-MPNN.*

We start by formally defining the logic $\mathsf{erFO}_{\mathrm{cnt}}^3$, which is a simple extension of $\mathsf{rFO}_{\mathrm{cnt}}^3$ from Section 5.2. Fix a set of relation types $R$ and a set of pair colors $\mathcal{C}$. Recall we consider knowledge graphs of the form $G = (V, E, R, \eta)$ where $\eta$ is a mapping assigning colors from $\mathcal{C}$ to pairs of nodes from $V$. The logic $\mathsf{erFO}_{\mathrm{cnt}}^3$ contains only binary formulas and it is defined inductively as follows: First, $a(x, y)$ for $a \in \mathcal{C}$, is in $\mathsf{erFO}_{\mathrm{cnt}}^3$. Second, if $\varphi(x, y)$ and $\psi(x, y)$ are in $\mathsf{erFO}_{\mathrm{cnt}}^3$, $N \geq 1$ is a positive integer, and $r \in R$, then the formulas

$$\neg\varphi(x, y), \quad \varphi(x, y) \wedge \psi(x, y), \quad \exists^{\geq N} z\, (\varphi(x, z) \wedge r(z, y)), \quad \exists^{\geq N} z\, (\varphi(x, z) \wedge \neg r(z, y))$$

are also in $\mathsf{erFO}_{\mathrm{cnt}}^3$. As expected, $a(u, v)$ holds in $G = (V, E, R, \eta)$ if $\eta(u, v) = a$, and $\exists^{\geq N} z\, (\varphi(u, z) \wedge \ell(z, v))$ holds in $G$, for $\ell \in R \cup \{\neg r \mid r \in R\}$, if there are at least $N$ nodes $w \in V$ for which $\varphi(u, w)$ and $\ell(w, v)$ hold in $G$.

Intuitively, $\mathsf{erFO}_{\mathrm{cnt}}^3$ extends $\mathsf{rFO}_{\mathrm{cnt}}^3$ with *negated modalities*, that is, one can check for non-neighbors of a node. As it turns out, $\mathsf{erFO}_{\mathrm{cnt}}^3$ is strictly more expressive than $\mathsf{rFO}_{\mathrm{cnt}}^3$.

**Proposition A.15.** *There is a formula in $\mathsf{erFO}_{\mathrm{cnt}}^3$ that cannot be expressed by any formula in $\mathsf{rFO}_{\mathrm{cnt}}^3$.*

*Proof.* Consider the following two knowledge graphs $G_1$ and $G_2$ over $R = \{r\}$ and colors $\mathcal{C} = \{\mathrm{Green}\}$. The graph $G_1$ has two nodes $u$ and $v$ and one edge $r(u, v)$. The graph $G_2$ has three nodes $u, v$ and $w$ and one edge $r(u, v)$ (hence $w$ is an isolated node). In both graphs, all pairs are colored Green.

We show first that for every formula $\varphi(x, y)$ in $\mathsf{rFO}_{\mathrm{cnt}}^3$, it is the case that $G_1, u, n \models \varphi$ iff $G_2, u, n \models \varphi$, for $n \in \{u, v\}$. We proceed by induction on the formula. For $\varphi(x, y) = \mathrm{Green}(x, y)$, we have

that $G_1, u, n \models \varphi$ and $G_2, u, n \models \varphi$ and hence we are done. Suppose now that $\varphi(x,y) = \neg\psi(x,y)$. Assume $G_1, u, n \models \varphi$ for $n \in \{u, v\}$. Then $G_1, u, n \not\models \psi$, and by inductive hypothesis, we have $G_2, u, n \not\models \psi$ and then $G_2, u, n \models \varphi$. The other direction is analogous. Assume now that $\varphi(x,y) = \psi_1(x,y) \wedge \psi_2(x,y)$. Suppose $G_1, u, n \models \varphi$ for $n \in \{u, v\}$. Then $G_1, u, n \models \psi_1$ and $G_1, u, n \models \psi_2$, and by inductive hypothesis, $G_2, u, n \models \psi_1$ and $G_2, u, n \models \psi_2$. Then $G_2, u, n \models \varphi$. The other direction is analogous. Finally, suppose that $\varphi(x,y) = \exists^{\geq N} z(\psi(x,z) \wedge r(z,y))$. Assume that $G_1, u, n \models \varphi$ for $n \in \{u, v\}$. Then there exist at least $N$ nodes $w \in \mathcal{N}_r(n)$ in $G_1$ such that $G_1, u, w \models \psi$. Since $\mathcal{N}_r(u) = \emptyset$, we have $n = v$. As $\mathcal{N}_r(v) = \{u\}$, we have $w = u$. Since the neighborhood $\mathcal{N}_r(v)$ is the same in $G_1$ and $G_2$, and by the inductive hypothesis, we have that there are at least $N$ nodes $w \in \mathcal{N}_r(v)$ in $G_2$ such that $G_2, u, w \models \psi$. This implies that $G_2, u, v \models \varphi$. The other direction is analogous.

Now consider the erFO$^3_{\text{cnt}}$ formula $\varphi(x,y) = \exists^{\geq 2} z(\text{Green}(x,z) \wedge \neg r(z,y))$. We claim that there is no rFO$^3_{\text{cnt}}$ formula equivalent to $\varphi(x,y)$. By contradiction, suppose we have such an equivalent formula $\varphi'$. As shown above, we have $G_1, u, v \models \varphi'$ iff $G_2, u, v \models \varphi'$. On the other hand, by definition, we have that $G_1, u, v \not\models \varphi$ and $G_2, u, v \models \varphi$. This is a contradiction. $\qquad\square$

Now we are ready to prove Theorem 5.3. We follow the same strategy as in the proof of the backward direction of Theorem A.11.

Let $\varphi(x,y)$ be a formula in erFO$^3_{\text{cnt}}$, for relation types $R$ and pair colors $\mathcal{C}$. Let $\varphi_1, \ldots, \varphi_L$ be an enumeration of the subformulas of $\varphi$ such that if $\varphi_i$ is a subformula of $\varphi_j$, then $i \leq j$. In particular, $\varphi_L = \varphi$. We will construct a C-MPNN $\mathcal{A}_\varphi$ with $L$ layers computing $L$-dimensional features in each layer. At layer $\ell \in \{1, \ldots, L\}$, the $\ell$-th component of the feature $\boldsymbol{h}^{(\ell)}_{v|u,q}$ will correspond to 1 if $\varphi_\ell$ is satisfied on $(u,v)$, and 0 otherwise. The query relation $q$ plays no role in the construction, that is, for any possible $q \in R$ the output of $\mathcal{A}_\varphi$ is the same. Hence, for simplicity, in the remaining of the proof we shall write $\boldsymbol{h}^{(t)}_{v|u}$ instead of $\boldsymbol{h}^{(t)}_{v|u,q}$. We add an additional final layer that simply outputs the last component of the feature vector.

We use models of C-MPNNs of the following form:

$$\boldsymbol{h}^{(t+1)}_{v|u} = \sigma\Big(\boldsymbol{W}_0 \boldsymbol{h}^{(t)}_v + \sum_{r \in R} \sum_{w \in \mathcal{N}_r(v)} \boldsymbol{W}_r \boldsymbol{h}^{(t)}_{w|u} + \boldsymbol{W}_1 \sum_{w \in V} \boldsymbol{h}^{(t)}_{w|u} + \boldsymbol{b}\Big),$$

where $\boldsymbol{W}_0, \boldsymbol{W}_1 \in \mathbb{R}^{L \times L}$ are parameter matrices and $\boldsymbol{b} \in \mathbb{R}^L$ is the bias term. As message function $\text{MSG}_r$ we use $\text{MSG}_r(\boldsymbol{h}) = \boldsymbol{W}_r \boldsymbol{h}$, where $\boldsymbol{W}_r \in \mathbb{R}^{L \times L}$ is a parameter matrix . For the non-linearity $\sigma$ we use the truncated ReLU function $\sigma(x) = \min(\max(0,x), 1)$. The $\ell$-th row of $\boldsymbol{W}_0$, $\boldsymbol{W}_1$ and $\boldsymbol{W}_r$, and the $\ell$-th entry of $\boldsymbol{b}$ are defined as follows (omitted entries are 0):

1. If $\varphi_\ell(x,y) = a(x,y)$ for a color $a \in \mathcal{C}$, then $(\boldsymbol{W}_0)_{\ell\ell} = 1$.
2. If $\varphi_\ell(x,y) = \neg\varphi_k(x,y)$ then $(\boldsymbol{W}_0)_{\ell k} = -1$, and $b_\ell = 1$.
3. If $\varphi_\ell(x,y) = \varphi_j(x,y) \wedge \varphi_k(x,y)$ then $(\boldsymbol{W}_0)_{\ell j} = 1$, $(\boldsymbol{W}_0)_{\ell k} = 1$ and $b_\ell = -1$.
4. If $\varphi_\ell(x,y) = \exists^{\geq N} z\,(\varphi_k(x,z) \wedge r(z,y))$ then $(\boldsymbol{W}_r)_{\ell k} = 1$ and $b_\ell = -N + 1$.
5. If $\varphi_\ell(x,y) = \exists^{\geq N} z\,(\varphi_k(x,z) \wedge \neg r(z,y))$ then $(\boldsymbol{W}_r)_{\ell k} = -1$, $(\boldsymbol{W}_1)_{\ell k} = 1$ and $b_\ell = -N + 1$.

Let $G = (V, E, R, \eta)$ be a knowledge graph with pair colors from $\mathcal{C}$. In order to apply $\mathcal{A}_\varphi$ to $G$, we choose the initialization INIT such that $L$-dimensional initial features $\boldsymbol{h}^{(0)}_{v|u}$ satisfy $(\boldsymbol{h}^{(0)}_{v|u})_\ell = 1$ if $\varphi_\ell = a(x,y)$ and $\eta(u,v) = a$ and $(\boldsymbol{h}^{(0)}_{v|u})_\ell = 0$ otherwise. That is, the $L$-dimensional initial feature $\boldsymbol{h}^{(0)}_{v|u}$ is a one-hot encoding of the color of $(u,v)$. Using the same arguments as in the proof of Theorem A.11, we have $(\boldsymbol{h}^{(t)}_{v|u})_\ell = 1$ if $G, u, v \models \varphi_\ell$ and $(\boldsymbol{h}^{(t)}_{v|u})_\ell = 0$ otherwise, for all $u, v \in V$ and $t \in \{\ell, \ldots, L\}$. In particular, after $L$ layers, $\mathcal{A}_\varphi$ calculates $\boldsymbol{h}^{(L)}_{v|u}$ such that $(\boldsymbol{h}^{(L)}_{v|u})_L = 1$ if $G, u, v \models \varphi$ and $(\boldsymbol{h}^{(L)}_{v|u})_L = 0$ otherwise. As layer $L+1$ extracts the $L$-th component of the feature vector, the result follows.

*Remark* A.16. We note that in Barceló et al. [3], it is shown that a more expressive graded modal logic (with more expressive modalities) can be captured by R-MPNNs in the context of single-relation

graphs. It is by no means obvious that an adaption of this logic to the multi-relational case is captured by our C-MPNNs. We leave as an open problem to find more expressive logics that can be captured by C-MPNNs.

## A.5   Proofs of the results from Section 5.3

First, recall the definition of $\mathsf{rawl}_2$. Given a knowledge graph $G = (V, E, R, c, \eta)$, we have

$$\mathsf{rawl}_2^{(0)}(u, v) = \eta(u, v),$$
$$\mathsf{rawl}_2^{(t+1)}(u, v) = \tau\big(\mathsf{rawl}_2^{(t)}(u, v), \{\!\!\{(\mathsf{rawl}_2^{(t)}(u, w), r) \mid w \in \mathcal{N}_r(v), r \in R)\}\!\!\}\big)$$

Note that $\mathsf{rawl}_2$, and hence C-MPNNs, are *one-directional*: the neighborhood $\mathcal{N}_r(v)$ only considers facts in one direction, in this case, *from* neighbors *to* $v$. Hence, a natural extension is to consider *bi-directional* neighborhoods. Fortunately, we can define this extension by simply applying the same test $\mathsf{rawl}_1$ to knowledge graphs extended with *inverse* relations. We formalize this below.

For a test $\mathsf{T}$, we sometimes write $\mathsf{T}(G, u, v)$, or $\mathsf{T}(G, v)$ in case of unary tests, to emphasize that the test is applied over $G$, and $\mathsf{T}(G)$ for the node/pairwise coloring given by the test. Let $G = (V, E, R, c, \eta)$ be a knowledge graph. We define its *augmented knowledge graph* to be $G^+ = (V, E^+, R^+, c, \eta)$, where $R^+$ is the disjoint union of $R$ and $\{r^- \mid r \in R\}$, and

$$E^+ = E \cup \{r^-(v, u) \mid r(u, v) \in E, u \neq v\}.$$

We define the *augmented relational asymmetric local 2-WL* test on $G$, denoted by $\mathsf{rawl}_2^+$, as

$$\mathsf{rawl}_2^{+\,(t)}(G, u, v) = \mathsf{rawl}_2^{(t)}(G^+, u, v),$$

for all $t \geq 0$ and $u, v \in V$. As we show below, $\mathsf{rawl}_2^+$ is strictly more powerful than $\mathsf{rawl}_2$.

**Proposition A.17.** *The following statements hold:*

1. *For all $t \geq 0$ and all knowledge graphs $G$, we have $\mathsf{rawl}_2^{+\,(t)}(G) \preceq \mathsf{rawl}_2^{(t)}(G)$.*

2. *There is a knowledge graph $G$ and nodes $u, v, u', v'$ such that $\mathsf{rawl}_2^{+\,(1)}(G, u, v) \neq \mathsf{rawl}_2^{+\,(1)}(G, u', v')$ but $\mathsf{rawl}_2^{(t)}(G, u, v) = \mathsf{rawl}_2^{(t)}(G, u', v')$ for all $t \geq 1$.*

*Proof.* To prove $\mathsf{rawl}_2^+(G) \preceq \mathsf{rawl}_2(G)$ first, we consider induction on iteration $t$. The base case for $t = 0$ is trivial by the assumption, so it is enough to consider the inductive step. We need to show that for some $k$,

$$\mathsf{rawl}_2^{+\,(k+1)}(u, v) = \mathsf{rawl}_2^{+\,(k+1)}(u', v') \implies \mathsf{rawl}_2^{(k+1)}(u, v) = \mathsf{rawl}_2^{(k+1)}(u', v')$$

By the definition of $\mathsf{rawl}_2^{+\,(k+1)}(u, v)$, $\mathsf{rawl}_2^{+\,(k+1)}(u', v')$ and the injectivity of $\tau$, it holds that

$$\mathsf{rawl}_2^{+\,(k)}(u, v) = \mathsf{rawl}_2^{+\,(k)}(u', v')$$

$$\{\!\!\{(\mathsf{rawl}_2^{+\,(k)}(u, x), r) \mid x \in \mathcal{N}_r^+(v), r \in R^+\}\!\!\} = \{\!\!\{(\mathsf{rawl}_2^{+\,(k)}(u', x'), r') \mid x' \in \mathcal{N}_{r'}^+(v'), r' \in R^+\}\!\!\}$$

Because all inverse relations $r^-$ are newly introduced, so it is impossible to be mixed with $r$, we can split the second equation into the following equations:

$$\{\!\!\{(\mathsf{rawl}_2^{+\,(k)}(u, x), r) \mid x \in \mathcal{N}_r(v), r \in R\}\!\!\} = \{\!\!\{(\mathsf{rawl}_2^{+\,(k)}(u', x'), r') \mid x' \in \mathcal{N}_{r'}(v'), r' \in R\}\!\!\}$$

$$\{\!\!\{(\mathsf{rawl}_2^{+\,(k)}(u, y), r^-) \mid y \in \mathcal{N}_{r^-}(v), r \in R\}\!\!\} = \{\!\!\{(\mathsf{rawl}_2^{+\,(k)}(u', y'), r'^-) \mid y' \in \mathcal{N}_{r'^-}(v'), r \in R\}\!\!\}$$

By the inductive hypothesis and unpacking the first equation, we can further imply that

$$\{\!\!\{(\mathsf{rawl}_2^{(k)}(u, x), r) \mid x \in \mathcal{N}_r(v), r \in R\}\!\!\} = \{\!\!\{(\mathsf{rawl}_2^{(k)}(u', x'), r') \mid x' \in \mathcal{N}_{r'}(v'), r' \in R\}\!\!\}$$

Thus, By definition of $\mathsf{rawl}_2^{(k+1)}(u, v)$, $\mathsf{rawl}_2^{(k+1)}(u', v')$ it holds that

$$\mathsf{rawl}_2^{(k+1)}(u, v) = \mathsf{rawl}_2^{(k+1)}(u', v')$$

For the counterexample, we consider a relational graph with two types of relation $G' = (V', E', R', c, \eta)$ such that $V' = \{u, v, v'\}$, $E' = \{r_1(v, u), r_2(v', u)\}$, and $R' = \{r_1, r_2\}$. Let the initial pairwise labeling $\eta$ for node pairs $(u, v)$ and $(u', v)$ satisfy $\mathsf{rawl}_2^{+(0)}(u, v) = \mathsf{rawl}_2^{+(0)}(u, v')$ and $\mathsf{rawl}_2^{(0)}(u, v) = \mathsf{rawl}_2^{(0)}(u, v')$. For such graph $G'$, we consider node pair $(u, v)$ and $(u, v')$. For $\mathsf{rawl}_2^{(t)}$ where $t \geq 0$, we show by induction that $\mathsf{rawl}_2^{(t)}(u, v) = \mathsf{rawl}_2^{(t)}(u, v')$. The base case is trivial by assumption. The inductive step shows that by the inductive hypothesis,

$$\mathsf{rawl}_2^{(t+1)}(u, v) = \tau(\mathsf{rawl}_2^{(t)}(u, v), \{\!\{\}\!\})$$
$$= \tau(\mathsf{rawl}_2^{(t)}(u, v'), \{\!\{\}\!\})$$
$$= \mathsf{rawl}_2^{(t+1)}(u, v')$$

On the other hand, we have

$$\mathsf{rawl}_2^{+(1)}(u, v) = \tau(\mathsf{rawl}_2^{+(0)}(u, v), \{\!\{(\mathsf{rawl}_2^{+(0)}(x, v), r_1^-)\}\!\})$$
$$\neq \mathsf{rawl}_2^{+(1)}(u, v') = \tau(\mathsf{rawl}_2^{+(0)}(u, v'), \{\!\{(\mathsf{rawl}_2^{+(0)}(x, v'), r_2^-))\}\!\})$$

$\square$

We can also extend C-MPNNs with bi-directionality in an obvious way, obtaining *augmented* C-MPNNs. By applying Theorem 5.1 to the augmented graph $G^+$, we obtain the equivalence between augmented C-MPNNs and the test $\mathsf{rawl}_2^+$ directly. In turn, Proposition A.17 implies that augmented C-MPNNs are strictly more powerful that C-MPNNs in distinguishing nodes in a graph.

Recall the definition of $\mathsf{rwl}_2$. Given a knowledge graph $G = (V, E, R, c, \eta)$, we have

$$\mathsf{rwl}_2^{(t+1)}(u, v) = \tau\big(\mathsf{rwl}_2^{(t)}(u, v), \{\!\{(\mathsf{rwl}_2^{(t)}(w, v), r) \mid w \in \mathcal{N}_r(u), r \in R)\}\!\},$$
$$\{\!\{(\mathsf{rwl}_2^{(t)}(u, w), r) \mid w \in \mathcal{N}_r(v), r \in R)\}\!\}\big)$$

We define the augmented version $\mathsf{rwl}_2^+$ in an obvious way, that is,

$$\mathsf{rwl}_2^{+(t)}(G, u, v) = \mathsf{rwl}_2^{(t)}(G^+, u, v),$$

for all $t \geq 0$ and $u, v \in V$.

We will drop the notation of $G$ during the proof when the context is clear for simplicity.

**Proposition A.18.** *For all $t \geq 0$ and all knowledge graph $G$, let $\mathsf{rwl}_2^{(0)}(G) \equiv \mathsf{rawl}_2^{(0)}(G)$. The following statements hold:*

1. *For every $t > 0$, it holds that $\mathsf{rwl}_2^{(t)}(G) \preceq \mathsf{rawl}_2^{(t)}(G)$.*

2. *There is a knowledge graph $G$ and pair of nodes $(u, v)$ and $(u', v')$ such that $\mathsf{rawl}_2^{(t)}(G, u, v) = \mathsf{rawl}_2^{(t)}(G, u', v')$ for all $t \geq 0$ but $\mathsf{rwl}_2^{(1)}(G, u, v) \neq \mathsf{rwl}_2^{(1)}(G, u', v')$.*

*Proof.* First, we show $\mathsf{rwl}_2^{(t)}(G) \preceq \mathsf{rawl}_2^{(t)}(G)$ and proceed by induction on $t$. The base case for $t = 0$ is trivial by assumption. For the inductive step, given that we have $\mathsf{rwl}_2^{(k+1)}(u, v) = \mathsf{rwl}_2^{(k+1)}(u', v')$, by the definition of $\mathsf{rwl}_2^{(k+1)}(u, v)$ and $\mathsf{rwl}_2^{(k+1)}(u', v')$, and as $\tau$ is injective, it holds that

$$\mathsf{rwl}_2^{(k)}(u, v) = \mathsf{rwl}_2^{(k)}(u', v')$$
$$\{\!\{(\mathsf{rwl}_2^{(k)}(x, v), r) \mid x \in \mathcal{N}_r(u), r \in R\}\!\} = \{\!\{(\mathsf{rwl}_2^{(k)}(x', v'), r') \mid x' \in \mathcal{N}_{r'}(u'), r' \in R\}\!\}$$
$$\{\!\{(\mathsf{rwl}_2^{(k)}(u, x), r) \mid x \in \mathcal{N}_r(v), r \in R\}\!\} = \{\!\{(\mathsf{rwl}_2^{(k)}(u', x'), r') \mid x' \in \mathcal{N}_{r'}(v'), r' \in R\}\!\}$$

Now, by the inductive hypothesis we have $\mathsf{rawl}_2^{(k)}(u, v) = \mathsf{rawl}_2^{(k)}(u', v')$. We can further transform the last equation by applying the inductive hypothesis again after unpacking the multiset. This results in

$$\{\!\{(\mathsf{rawl}_2^{(k)}(u, x), r) \mid x \in \mathcal{N}_r(v), r \in R\}\!\} = \{\!\{(\mathsf{rawl}_2^{(k)}(u', x'), r') \mid x' \in \mathcal{N}_{r'}(v'), r' \in R\}\!\}$$

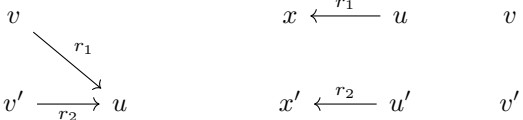

$$x \xrightarrow{\;r\;} u'$$

Figure 3: Graph $G'$ as the counter-example in Proposition A.18. It is also shown in Proposition A.20 to prove $\mathsf{rawl_2^{+}}^{(t)}(u,v) = \mathsf{rawl_2^{+}}^{(t)}(u',v')$ for all $t \geq 0$ but $\mathsf{rwl_2^{(1)}}(u,v) \neq \mathsf{rwl_2^{(1)}}(u',v')$.

$$v \qquad\qquad\qquad x \xleftarrow{\;r_1\;} u \qquad v$$
$$\searrow {\scriptstyle r_1}$$
$$v' \xrightarrow{\;r_2\;} u \qquad\qquad x' \xleftarrow{\;r_2\;} u' \qquad v'$$

Figure 4: Two counterexamples shown in Proposition A.17 and A.19. The left graph $G'$ is to show $\mathsf{rawl_2^{(t)}}(u,v) = \mathsf{rawl_2^{+}}^{(t)}(u',v')$ for all $t \geq 0$ but $\mathsf{rawl_2^{(1)}}(u,v) \neq \mathsf{rawl_2^{+}}^{(1)}(u',v')$, whereas the right graph $G''$ is to show $\mathsf{rwl_2^{(t)}}(u,v) = \mathsf{rwl_2^{(t)}}(u',v')$ for all $t \geq 0$ but $\mathsf{rwl_2^{+}}^{(1)}(u,v) \neq \mathsf{rwl_2^{+}}^{(1)}(u',v')$.

Thus, it holds that $\mathsf{rawl_2^{(k+1)}}(u,v) = \mathsf{rawl_2^{(k+1)}}(u',v')$ by definition of $\mathsf{rawl_2^{(k+1)}}(u,v)$ and $\mathsf{rawl_2^{(k+1)}}(u',v')$.

For counter-example, we show the case for $t \geq 0$. Consider a relational graph $G' = (V', E', R, c, \eta)$ such that $V' = \{u, u', v, x\}$, $E' = \{r(x, u')\}$, and $R = \{r\}$ with the initial labeling $\eta$ for node pairs $(u,v)$ and $(u',v)$ satisfies $\mathsf{rwl_2^{(0)}}(u,v) = \mathsf{rwl_2^{(0)}}(u',v)$ and $\mathsf{rawl_2^{(0)}}(u,v) = \mathsf{rawl_2^{(0)}}(u',v)$. For such graph $G'$, we consider node pair $(u,v)$ and $(u',v)$. For $\mathsf{rawl_2^{(t)}}$ where $t \geq 0$, we show by induction that $\mathsf{rawl_2^{(t)}}(u,v) = \mathsf{rawl_2^{(t)}}(u',v)$. The base case is trivial by assumption. The inductive step shows that by the inductive hypothesis:

$$\mathsf{rawl_2^{(t+1)}}(u,v) = \tau(\mathsf{rawl_2^{(t)}}(u,v), \{\!\!\{\}\!\!\})$$
$$= \tau(\mathsf{rawl_2^{(t)}}(u',v), \{\!\!\{\}\!\!\})$$
$$= \mathsf{rawl_2^{(t+1)}}(u',v)$$

On the other hand, we have

$$\mathsf{rwl_2^{(1)}}(u,v) = \tau(\mathsf{rwl_2^{(0)}}(u,v), \{\!\!\{\}\!\!\}, \{\!\!\{\}\!\!\})$$
$$\neq \mathsf{rwl_2^{(1)}}(u,v') = \tau(\mathsf{rwl_2^{(0)}}(u,v'), \{\!\!\{(\mathsf{rwl_2^{(0)}}(x,v'), r))\}\!\!\}, \{\!\!\{\}\!\!\})$$

$\square$

**Proposition A.19.** *For all $t \geq 0$ and all knowledge graph $G$, let $\mathsf{rwl_2^{+}}^{(0)}(G) \equiv \mathsf{rwl_2^{(0)}}(G)$. The following statements hold:*

1. *For every $t > 0$, $\mathsf{rwl_2^{+}}^{(t)}(G) \preceq \mathsf{rwl_2^{(t)}}(G)$*

2. *There is a knowledge graph $G''$ and pair of nodes $(u,v)$ and $(u',v')$ such that $\mathsf{rwl_2^{(t)}}(G'',u,v) = \mathsf{rwl_2^{(t)}}(G'',u',v')$ for all $t \geq 0$ but $\mathsf{rwl_2^{+}}^{(1)}(G'',u,v) \neq \mathsf{rwl_2^{+}}^{(1)}(G'',u',v')$.*

*Proof.* As before, we first prove $\mathsf{rwl_2^{+}}(G) \preceq \mathsf{rwl_2}(G)$ by induction on iteration $t$. The base case for $t = 0$ is trivial by the assumption. By the inductive hypothesis,

$$\mathsf{rwl_2^{+}}^{(k)}(u,v) = \mathsf{rwl_2^{+}}^{(k)}(u',v') \implies \mathsf{rwl_2^{(k)}}(u,v) = \mathsf{rwl_2^{(k)}}(u',v')$$

for some $k$. Thus, assuming $\mathsf{rwl}_2^{+(k+1)}(u,v) = \mathsf{rwl}_2^{+(k+1)}(u',v')$, by the definition of $\mathsf{rwl}_2^{+(k+1)}(u,v)$ and $\mathsf{rwl}_2^{+(k+1)}(u',v')$ and by the injectivity of $\tau$, it holds that

$$\mathsf{rwl}_2^{+(k)}(u,v) = \mathsf{rwl}_2^{+(k)}(u',v')$$

$$\{\!\{(\mathsf{rwl}_2^{+(k)}(x,v),r) \mid x \in \mathcal{N}_r^+(u), r \in R^+\}\!\} = \{\!\{(\mathsf{rwl}_2^{+(k)}(x',v'),r') \mid x' \in \mathcal{N}_{r'}^+(u'), r' \in R^+\}\!\}$$

$$\{\!\{(\mathsf{rwl}_2^{+(k)}(u,x),r) \mid x \in \mathcal{N}_r^+(v), r \in R^+\}\!\} = \{\!\{(\mathsf{rwl}_2^{+(k)}(u',x'),r') \mid x' \in \mathcal{N}_{r'}^+(v'), r' \in R^+\}\!\}$$

By the similar argument in proving $\mathsf{rwl}_2^+ \preceq \mathsf{rwl}_2$, we can split the multiset into two equations by decomposing $\mathcal{N}_r^+(u) = \mathcal{N}_r(u) \cup \mathcal{N}_{r^-}(u)$ and $\mathcal{N}_r^+(v) = \mathcal{N}_r(v) \cup \mathcal{N}_{r^-}(v)$. Thus, we have

$$\{\!\{(\mathsf{rwl}_2^{+(k)}(x,v),r) \mid x \in \mathcal{N}_r(u), r \in R\}\!\} = \{\!\{(\mathsf{rwl}_2^{+(k)}(x',v'),r') \mid x' \in \mathcal{N}_{r'}(u'), r' \in R\}\!\}$$

$$\{\!\{(\mathsf{rwl}_2^{+(k)}(y,v),r^-) \mid y \in \mathcal{N}_{r^-}(u), r \in R\}\!\} = \{\!\{(\mathsf{rwl}_2^{+(k)}(y',v'),r'^-) \mid y' \in \mathcal{N}_{r'^-}(u'), r' \in R\}\!\}$$

$$\{\!\{(\mathsf{rwl}_2^{+(k)}(u,x),r) \mid x \in \mathcal{N}_r(v), r \in R\}\!\} = \{\!\{(\mathsf{rwl}_2^{+(k)}(u',x'),r') \mid x' \in \mathcal{N}_{r'}(v'), r' \in R\}\!\}$$

$$\{\!\{(\mathsf{rwl}_2^{+(k)}(u,y),r^-) \mid y \in \mathcal{N}_{r^-}(v), r \in R\}\!\} = \{\!\{(\mathsf{rwl}_2^{+(k)}(u',y'),r'^-) \mid y' \in \mathcal{N}_{r'^-}(v'), r' \in R\}\!\}$$

By the inductive hypothesis and unpacking the first and the third equations, we can further imply that

$$\mathsf{rwl}_2^{(k)}(u,v) = \mathsf{rwl}_2^{(k)}(u',v')$$

$$\{\!\{(\mathsf{rwl}_2^{(k)}(x,v),r) \mid x \in \mathcal{N}_r(u), r \in R\}\!\} = \{\!\{(\mathsf{rwl}_2^{(k)}(x',v'),r') \mid x' \in \mathcal{N}_{r'}(u'), r' \in R\}\!\}$$

$$\{\!\{(\mathsf{rwl}_2^{(k)}(u,x),r) \mid x \in \mathcal{N}_r(v), r \in R\}\!\} = \{\!\{(\mathsf{rwl}_2^{(k)}(u',x'),r') \mid x' \in \mathcal{N}_{r'}(v'), r' \in R\}\!\}$$

This would results in $\mathsf{rwl}_2^{(k+1)}(u,v) = \mathsf{rwl}_2^{(k+1)}(u',v')$ by the definition of $\mathsf{rwl}_2^{(k+1)}(u,v)$ and $\mathsf{rwl}_2^{(k+1)}(u',v')$

For the counter-example, we consider a relational graph $G'' = (V'', E'', R'', c, \eta)$ such that $V'' = \{u,u',v,v',x,x'\}$, $E'' = \{r_1(u,x), r_2(u',x')\}$, and $R'' = \{r_1, r_2\}$ . For such graph $G''$, we set the initial labeling $\eta$ for node pairs $(u,v)$ and $(u',v')$ to satisfy $\mathsf{rwl}_2^{+(0)}(u,v) = \mathsf{rwl}_2^{+(0)}(u',v')$ and $\mathsf{rwl}_2^{(0)}(u,v) = \mathsf{rwl}_2^{(0)}(u',v')$. For such graph $G''$, we consider node pair $(u,v)$ and $(u',v')$.

Consider $\mathsf{rwl}_2^{(t)}$ where $t \geq 0$, we show by induction that $\mathsf{rwl}_2^{(t)}(u,v) = \mathsf{rwl}_2^{(t)}(u',v')$. The base case is trivial by assumption. The inductive step shows that by inductive hypothesis,

$$\mathsf{rwl}_2^{(t+1)}(u,v) = \tau(\mathsf{rwl}_2^{(t)}(u,v), \{\!\{\}\!\}, \{\!\{\}\!\})$$
$$= \tau(\mathsf{rwl}_2^{(t)}(u',v'), \{\!\{\}\!\}, \{\!\{\}\!\})$$
$$= \mathsf{rwl}_2^{(t+1)}(u',v')$$

On the other hand, we have

$$\mathsf{rwl}_2^{+(1)}(u,v) = \tau(\mathsf{rwl}_2^{+(0)}(u,v), \{\!\{(\mathsf{rwl}_2^{+(0)}(x,v),r_1^-)\}\!\}, \{\!\{\}\!\})$$
$$\neq \mathsf{rwl}_2^{+(1)}(u',v') = \tau(\mathsf{rwl}_2^{+(0)}(u',v'), \{\!\{(\mathsf{rwl}_2^{+(0)}(x',v'),r_2^-))\}\!\}, \{\!\{\}\!\})$$

$\square$

**Proposition A.20.** *For all $t \geq 0$ and all knowledge graph $G$, let $\mathsf{rwl}_2^{(0)}(G) \equiv \mathsf{rawl}_2^{+(0)}(G)$, then the following statement holds:*

1. *There is a knowledge graph $G$ and pair of nodes $(u,v)$ and $(u',v')$ such that $\mathsf{rwl}_2^{(t)}(G,u,v) = \mathsf{rwl}_2^{(t)}(G,u',v')$ for all $t \geq 0$ but $\mathsf{rawl}_2^{+(1)}(G,u,v) \neq \mathsf{rawl}_2^{+(1)}(G,u',v')$.*

2. *There is a knowledge graph $G'$ and pair of nodes $(u,v)$ and $(u',v')$ such that $\mathsf{rawl}_2^{+(t)}(G',u,v) = \mathsf{rawl}_2^{+(t)}(G',u',v')$ for all $t \geq 0$ but $\mathsf{rwl}_2^{(1)}(G',u,v) \neq \mathsf{rwl}_2^{(1)}(G',u',v')$.*

*Proof.* To show $\mathsf{rwl}_2(G)$ does not refine $\mathsf{rawl}_2^+(G)$, we consider a relational graph $G = (V, E, R, c, \eta)$ such that $V = \{u,u',v,v',x,x'\}$, $E = \{r_1(v,x), r_2(v',x')\}$, and $R = \{r_1, r_2\}$ . We let the initial

$$u \qquad v \xrightarrow{\quad r_1 \quad} x$$

$$u' \qquad v' \xrightarrow{\quad r_2 \quad} x'$$

Figure 5: Counterexample shown in Proposition A.20 to prove $\mathsf{rwl_2}^{(t)}(u,v) = \mathsf{rwl_2}^{(t)}(u',v')$ for all $t \geq 0$ but $\mathsf{rawl}_2^{+\,(1)}(u,v) \neq \mathsf{rawl}_2^{+\,(1)}(u',v')$.

labeling $\eta$ for node pairs $(u,v)$ and $(u',v')$ to satisfy $\mathsf{rwl}_2^{(0)}(u,v) = \mathsf{rwl}_2^{(0)}(u',v')$ and $\mathsf{rwl}_2^{(0)}(u,v) = \mathsf{rwl}_2^{(0)}(u',v')$. For such graph $G$, we consider node pair $(u,v)$ and $(u',v')$. For $\mathsf{rwl}_2^{(t)}$ where $t \geq 0$, we show by induction that $\mathsf{rwl}_2^{(t)}(u,v) = \mathsf{rwl}_2^{(t)}(u',v')$. The base case is trivial by assumption. The inductive step shows that by the inductive hypothesis,

$$\mathsf{rwl}_2^{(t+1)}(u,v) = \tau(\mathsf{rwl}_2^{(t)}(u,v), \{\!\{\}\!\}, \{\!\{\}\!\})$$
$$= \tau(\mathsf{rwl}_2^{(t)}(u',v'), \{\!\{\}\!\}, \{\!\{\}\!\})$$
$$= \mathsf{rwl}_2^{(t+1)}(u',v')$$

On the other hand, we have

$$\mathsf{rawl}_2^{+\,(1)}(u,v) = \tau(\mathsf{rawl}_2^{+\,(0)}(u,v), \{\!\{(\mathsf{rawl}_2^{+\,(0)}(u,x), r_1^-)\}\!\})$$
$$\neq \mathsf{rawl}_2^{+\,(1)}(u,v') = \tau(\mathsf{rawl}_2^{+\,(0)}(u',v'), \{\!\{(\mathsf{rawl}_2^{+\,(0)}(u',x'), r_2^-)\}\!\})$$

Finally, to show $\mathsf{rawl}_2^+$ does not refine $\mathsf{rwl}_2$, we demonstrate the case for $t \geq 0$. Consider a relational graph $G' = (V', E', R', c, \eta)$ such that $V' = \{u, u', v, x\}$, $E' = \{r(x, u')\}$, and $R' = \{r\}$. We let the initial labeling $\eta$ for node pairs $(u,v)$ and $(u',v)$ satisfy $\mathsf{rwl}_2^{(0)}(u,v) = \mathsf{rwl}_2^{(0)}(u',v)$ and $\mathsf{rawl}_2^{+\,(0)}(u,v) = \mathsf{rawl}_2^{+\,(0)}(u',v)$. For such graph $G'$, we consider node pair $(u,v)$ and $(u',v)$. For $\mathsf{rawl}_2^{+\,(t)}$ where $t \geq 0$, and show by induction that $\mathsf{rawl}_2^{+\,(t)}(u,v) = \mathsf{rawl}_2^{+\,(t)}(u',v)$. The base case is trivial by assumption. The inductive step shows that by the inductive hypothesis,

$$\mathsf{rawl}_2^{+\,(t+1)}(u,v) = \tau(\mathsf{rawl}_2^{+\,(t)}(u,v), \{\!\{\}\!\})$$
$$= \tau(\mathsf{rawl}_2^{+\,(t)}(u',v), \{\!\{\}\!\})$$
$$= \mathsf{rawl}_2^{+\,(t+1)}(u',v)$$

On the other hand, we have

$$\mathsf{rwl}_2^{(1)}(u,v) = \tau(\mathsf{rwl}_2^{(0)}(u,v), \{\!\{\}\!\}, \{\!\{\}\!\})$$
$$\neq \mathsf{rwl}_2^{(1)}(u,v') = \tau(\mathsf{rwl}_2^{(0)}(u,v'), \{\!\{(\mathsf{rwl}_2^{(0)}(x,v'), r))\}\!\}, \{\!\{\}\!\})$$

Note that the counter-example graph $G'$ here is identical to the one in Proposition A.18. $\qquad\square$

## B   Runtime analysis

In this section, we present the asymptotic time complexity of R-MPNN, C-MPNN, and a labeling-trick-based relation prediction model GraIL [30] in Table 3, taken from Zhu et al. [40]. We also consider 2-RN, the neural architecture from Barceló et al. [4] corresponding to $\mathsf{rwl}_2$ to showcase an example of higher-order GNNs. We present both the complexity of a single forward pass of the model as well as the amortized complexity of a single query to better reflect the practical use cases and the advantages stemming from parallelizability. To avoid confusion, we take CompGCN [32] as the example of R-MPNN, and *basic* C-MPNN, since specific architecture choices will impact the asymptotic complexity of the model.

**Notation.** Given a knowledge graph $G = (V, E, R, c)$, we have that $|V|, |E|, |R|$ represents the size of vertices, edges, and relation types, respectively. $d$ is the hidden dimension of the model, and $T$ is the number of layers in the model.

Table 3: Model asymptotic runtime complexities. Observe that the amortized complexity of 2-RN becomes obsolete by the fact that its forward pass being prohibitive in practice (quadratic in $|V|$).

| Model | Complexity of a forward pass | Amortized complexity of a query |
|---|---|---|
| R-MPNNs | $\mathcal{O}(T(|E|d + |V|d^2))$ | $\mathcal{O}(T(\frac{|E|d}{|R||V|^2} + \frac{d^2}{|R||V|} + d))$ |
| C-MPNNs | $\mathcal{O}(T(|E|d + |V|d^2))$ | $\mathcal{O}(T(\frac{|E|d}{|V|} + d^2))$ |
| GraIL | $\mathcal{O}(T(|E|d^2 + |V|d^2))$ | $\mathcal{O}(T(|E|d^2 + |V|d^2))$ |
| 2-RN | $\mathcal{O}(T(|V||E|d + |V|^2d^2))$ | $\mathcal{O}(T(\frac{|E|d}{|R||V|} + \frac{d^2}{|R|} + d))$ |

**Runtime of C-MPNNs.** Let us note that C-MPNNs subsume NBFNets. Indeed, if we consider C-MPNNs without readout and set the history function as $f(t) = 0$, we obtain (a slight generalization of) NBFNets, as stated in the paper. In terms of the runtime, C-MPNNs and NBFNets are comparable: even though the readout component of C-MPNNs incurs a linear overhead, this is dominated by other factors. Thus, as shown in Lemma 2 of Zhu et al. [40], the total complexity for a forward pass is $\mathcal{O}(T(|E|d + |V|d^2))$ to compute $|V|$ queries at the same time, resulting an amortized complexity of $\mathcal{O}(T(\frac{|E|d}{|V|} + d^2))$. See Zhu et al. [40] for detailed discussion and proof.

**Comparison with R-MPNNs.** As directly carrying out relational message passing in R-MPNN for each triplet is costly, a common way to carry out link prediction task is to first compute all node representations and then use a binary decoder to perform link prediction. This would result in $\mathcal{O}(T(|E|d + |V|d^2))$ for a complete forward pass. Consequently, we obtain $|R||V|^2$ triplets, and since passing through binary decoder has a complexity of $\mathcal{O}(d)$, we have that the amortized complexity for each query is $\mathcal{O}(T(\frac{|E|d}{|R||V|^2} + \frac{d^2}{|R||V|} + d))$. Although the complexity of a forward pass for R-MPNN and C-MPNN are the same, we compute $|R||V|^2$ queries with R-MPNN but only $|V|$ queries with C-MPNN, resulting difference in amortized complexity of a single query. However, as discussed in Section 4, this comes at the cost of expressivity.

**Comparison with the architectures using labeling trick.** GraIL is an architecture using labeling trick and we focus on this to make the comparison concrete. In terms of runtime, architectures that rely on the labeling trick are typically slower, since they need to label both the source and the target node, as we state in Section 4. This yields worse runtime complexity for these models, and particularly for GraIL [4], with an amortized complexity of a query being the same as the one for a forward pass, that is, $\mathcal{O}(T(|E|d^2 + |V|d^2))$. In C-MPNNs, we only label the source node $u$, which allows parallel comparison for all queries in the form $q(u, ?)$: a single forward pass would compute all hidden representations of these queries. With the labeling trick, $|V|$ forward passes need to be carried out as each time, both source $u$ and target $v$ need to be specified. See Zhu et al. [40] for detailed discussion and proof.

**Comparison with higher-order GNNs.** Regarding higher-order GNNs, we take 2-RN, the neural architecture from Barceló et al. [4] corresponding to $\text{rwl}_2$. These higher-order models require $\mathcal{O}((|V||E|d + |V|^2d^2))$ in each layer (assuming each message function takes $\mathcal{O}(d)$ as before) to update all pairwise representations in a single forward pass, which is computationally prohibitive in practice. This makes the study of the amortized complexity obsolete, but we provide an analysis for the sake of completeness. Similar to R-MPNN, we need an additional query-specific unary decoder for the task of link prediction on knowledge graphs, resulting in $\mathcal{O}(d)$ complexity overhead. The amortized complexity of a single query is thus $\mathcal{O}(T(\frac{|E|d}{|R||V|} + \frac{d^2}{|R|} + d))$.

## C Experiments on inductive link prediction

### C.1 Details of the experiments reported in Section 6

In this section, we report the details of the experiments reported in the body of this paper. Specifically, we report the performance of some baseline models (Table 4), dataset statistics (Table 5), hyperparameters (Table 6), and the number of trainable parameters for each model variation (Table 7).

---

[4]Comparing to C-MPNN, the complexity for $\text{MSG}_r$ is different because GraIL utilizes RGCN as relational message passing model, which needs $\mathcal{O}(d^2)$ for linear transformation in its $\text{MSG}_r$.

Table 4: Inductive relation prediction results. We use *basic* C-MPNN architecture with $\textsc{Agg} = \text{sum}$, $\textsc{Msg} = \textsc{Msg}_r^1$, and $\textsc{Init} = \textsc{Init}^2$ with no readout component. All the baselines are taken from the respective works.

| Model | WN18RR | | | | FB15k-237 | | | |
|---|---|---|---|---|---|---|---|---|
| | v1 | v2 | v3 | v4 | v1 | v2 | v3 | v4 |
| NeuralLP[35] | 0.744 | 0.689 | 0.462 | 0.671 | 0.529 | 0.589 | 0.529 | 0.559 |
| DRUM [25] | 0.744 | 0.689 | 0.462 | 0.671 | 0.529 | 0.587 | 0.529 | 0.559 |
| RuleN [20] | 0.809 | 0.782 | 0.534 | 0.716 | 0.498 | 0.778 | 0.877 | 0.856 |
| GraIL [30] | 0.825 | 0.787 | 0.584 | 0.734 | 0.642 | 0.818 | 0.828 | 0.893 |
| C-MPNN | **0.932** | **0.896** | **0.900** | **0.881** | **0.794** | **0.906** | **0.947** | **0.933** |

Table 5: Dataset statistics for the inductive relation prediction experiments. **#Query\*** is the number of queries used in the validation set. In the training set, all triplets are used as queries.

| Dataset | | #Relation | Train & Validation | | | Test | | |
|---|---|---|---|---|---|---|---|---|
| | | | #Nodes | #Triplet | #Query* | #Nodes | #Triplet | #Query |
| WN18RR | $v_1$ | 9 | 2,746 | 5,410 | 630 | 922 | 1,618 | 188 |
| | $v_2$ | 10 | 6,954 | 15,262 | 1,838 | 2,757 | 4,011 | 441 |
| | $v_3$ | 11 | 12,078 | 25,901 | 3,097 | 5,084 | 6,327 | 605 |
| | $v_4$ | 9 | 3,861 | 7,940 | 934 | 7,084 | 12,334 | 1,429 |
| FB15k-237 | $v_1$ | 180 | 1,594 | 4,245 | 489 | 1,093 | 1,993 | 205 |
| | $v_2$ | 200 | 2,608 | 9,739 | 1,166 | 1,660 | 4,145 | 478 |
| | $v_3$ | 215 | 3,668 | 17,986 | 2,194 | 2,501 | 7,406 | 865 |
| | $v_4$ | 219 | 4,707 | 27,203 | 3,352 | 3,051 | 11,714 | 1,424 |

Table 6: Hyperparameters for inductive experiments with C-MPNN.

| Hyperparameter | | WN18RR | FB15k-237 |
|---|---|---|---|
| **GNN Layer** | Depth($T$) | 6 | 6 |
| | Hidden Dimension | 32 | 32 |
| **Decoder Layer** | Depth | 2 | 2 |
| | Hidden Dimension | 64 | 64 |
| **Optimization** | Optimizer | Adam | Adam |
| | Learning Rate | 5e-3 | 5e-3 |
| **Learning** | Batch size | 8 | 8 |
| | #Negative Samples | 32 | 32 |
| | Epoch | 20 | 20 |
| | Adversarial Temperature | 1 | 1 |

Table 7: Number of trainable parameters used in inductive relation prediction experiments for C-MPNN architectures with $\textsc{Init}^2$ initialization .

| Model architectures | | WN18RR | | | | FB15k-237 | | | |
|---|---|---|---|---|---|---|---|---|---|
| Agg | Msg$_r$ | v1 | v2 | v3 | v4 | v1 | v2 | v3 | v4 |
| sum | $\textsc{Msg}_r^1$ | 132k | 144k | 157k | 132k | 2,310k | 2,564k | 2,755k | 2,806k |
| sum | $\textsc{Msg}_r^2$ | 21k | 22k | 22k | 21k | 98k | 107k | 113k | 115k |
| sum | $\textsc{Msg}_r^3$ | 128k | 141k | 153k | 128k | 2,240k | 2,488k | 2,673k | 2,722k |
| PNA | $\textsc{Msg}_r^1$ | 199k | 212k | 225k | 199k | 2,377k | 2,632k | 2,823k | 2,874k |
| PNA | $\textsc{Msg}_r^2$ | 89k | 89k | 90k | 89k | 165k | 174k | 181k | 183k |
| PNA | $\textsc{Msg}_r^3$ | 196k | 208k | 221k | 196k | 2,308k | 2,555k | 2,740k | 2,790k |

All of the model variations minimize the negative log-likelihood of positive and negative facts. We follow the *partial completeness assumption* [10] by randomly corrupting the head entity or the tail entity to generate the negative samples. We parameterize the conditional probability of a fact $q(u,v)$ by $p(v \mid u, q) = \sigma(f(\boldsymbol{h}_{v|u,q}^{(T)}))$, where $\sigma$ is the sigmoid function and $f$ is 2-layer MLP. Following RotatE [29], we adopt *self-adversarial negative sampling* by sampling negative triples from the following distribution with $\alpha$ as the *adversarial temperature*:

$$\mathcal{L}(v \mid u, q) = -\log p(v \mid u, q) - \sum_{i=1}^{k} w_{i,\alpha} \log(1 - p(v_i' \mid u_i', q))$$

where $k$ is the number of negative samples for one positive sample and $(u_i', q, v_i')$ is the $i$-th negative sample. Finally, $w_i$ is the weight for the $i$-th negative sample, given by

$$w_{i,\alpha} := \mathrm{Softmax}\left(\frac{\log(1 - p(v_i' \mid u_i', q))}{\alpha}\right).$$

### C.2   Experiments for evaluating the effect of initialization functions

**Initialization functions (Q3).** We argued that the initialization function $\mathrm{INIT}(u, v, q)$ needs to satisfy the property of *target node distinguishability* to compute binary invariants. To validate the impact of different initialization regimes, we conduct a further experiment which is reported in Table 8, with the same experiment settings as in Section 6. In addition to the initialization functions $\mathrm{INIT}^1$, $\mathrm{INIT}^2$, $\mathrm{INIT}^3$ defined in Section 4.1, we also experiment with a simple function $\mathrm{INIT}^0$ which assigns $\boldsymbol{0}$ to all nodes. As expected, using $\mathrm{INIT}^0$ initialization, results in a very sharp decrease in model performance in WN18RR, but less so in FB15k-237. Intuitively, the model suffers more in WN18RR since there are much fewer relations, and it is harder to distinguish node pairs without an initialization designed to achieve this. Perhaps one of the simplest functions satisfying target node distinguishability criteria is $\mathrm{INIT}^1 = \mathbb{1}_{u=v} * \boldsymbol{1}$, which pays no respect to the target query relation. Empirically, $\mathrm{INIT}^1$ achieves strong results, showing that even the simplest function ensuring this property could boost the model performance. Interestingly, the performance of models using $\mathrm{INIT}^1$ match or exceed models using $\mathrm{INIT}^2$, even though the latter additionally has a relation-specific learnable query vector. Note, however, that this shall not undermine the role of the learnable query relation: integrating the learnable query vector either in the initialization function *or* in the message computation function seems to suffice.

Table 8: Inductive relation prediction of C-MPNN using $\mathrm{AGG} = \mathrm{sum}$, $\mathrm{MSG}_r = \mathrm{MSG}_r^1$, $f(t) = t$ and different initialization methods.

| Initialization | WN18RR | | | | FB15k-237 | | | |
|---|---|---|---|---|---|---|---|---|
| $\mathrm{INIT}(u,v,q)$ | v1 | v2 | v3 | v4 | v1 | v2 | v3 | v4 |
| $\mathrm{INIT}^0(u,v,q)$ | 0.615 | 0.715 | 0.811 | 0.654 | 0.777 | 0.903 | 0.894 | 0.910 |
| $\mathrm{INIT}^1(u,v,q)$ | 0.932 | 0.894 | **0.902** | **0.883** | **0.809** | **0.927** | 0.944 | 0.911 |
| $\mathrm{INIT}^2(u,v,q)$ | 0.932 | **0.896** | 0.900 | 0.881 | 0.794 | 0.906 | **0.947** | 0.933 |
| $\mathrm{INIT}^3(u,v,q)$ | **0.934** | 0.890 | 0.894 | 0.877 | 0.804 | 0.924 | 0.941 | **0.944** |

**Random initialization (Q3).** The idea of random node initialization is known to lead to more expressive models [26, 1]. Inspired by this, we can incorporate varying degrees of randomization to the initialization, satisfying the target node distinguishability property in expectation. The key advantage is that the resulting models are still inductive and achieve a nontrivial expressiveness gain over alternatives. In this case, the models with $\mathrm{INIT}^3$ perform closer to the models with $\mathrm{INIT}^2$, but we do not see a particular advantage on these benchmarks.

## D   Experiments on transductive link prediction

We further conducted transductive link prediction experiments to empirically validate that C-MPNNs are more expressive than R-MPNNs via the abstraction given by $\mathrm{rawl}_2$ and $\mathrm{rwl}_1$, respectively **(Q4)**.

Table 9: Transductive knowledge graph completion task of C-MPNNs and R-MPNNs on WN18RR and FB15k-237. Here, C-MPNN-*basic* refers to *basic* C-MPNN model without readout.

| Model class | Architectures | WN18RR | | | FB15k-237 | | |
| --- | --- | --- | --- | --- | --- | --- | --- |
| | | MR | MRR | Hits@10 | MR | MRR | Hits@10 |
| **R-MPNNs** | RGCN | 3069 | 0.367 | 0.405 | 210 | 0.205 | 0.387 |
| | CompGCN | 3590 | 0.433 | 0.519 | 217 | 0.334 | 0.514 |
| **C-MPNNs** | NeuralLP | – | 0.435 | 0.566 | – | 0.240 | 0.362 |
| | DRUM | – | 0.486 | 0.586 | – | 0.343 | 0.516 |
| | C-MPNN-*basic* | **687** | **0.534** | **0.643** | **121** | **0.400** | **0.583** |

Table 10: Dataset statistics for transductive experiments on WN18RR and FB15k-237.

| Dataset | #Nodes | #Relation | #Triplet | | |
| --- | --- | --- | --- | --- | --- |
| | | | #Train | #Valid | #Test |
| WN18RR | 40,943 | 11 | 86,835 | 3,034 | 3,134 |
| FB15k-237 | 14,541 | 237 | 272,115 | 17,535 | 20,466 |

Table 11: Hyperparameters for transductive experiments on FB15k-237 and WN18RR with C-MPNN.

| Hyperparameter | | WN18RR | FB15k-237 |
| --- | --- | --- | --- |
| **GNN Layer** | Depth($T$) | 6 | 6 |
| | Hidden Dimension | 32 | 32 |
| **Decoder Layer** | Depth | 2 | 2 |
| | Hidden Dimension | 64 | 64 |
| **Optimization** | Optimizer | Adam | Adam |
| | Learning Rate | 5e-3 | 5e-3 |
| **Learning** | Batch size | 8 | 8 |
| | #Negative Sample | 32 | 32 |
| | Epoch | 20 | 20 |
| | Adversarial Temperature | 1 | 1 |

### D.1 Experiments on WN18RR and FB15k-237

**Datasets.** We use two benchmark datasets for transductive link prediction experiments, namely WN18RR [31] and FB15k-237 [9], with the provided standardized train-test split. Similar to the inductive relation prediction experiments, we augment the fact $r(u, v)$ with its inverse fact $r^{-1}(v, u)$. The detailed data statistics are shown in Table 10.

**Implementation.** For *basic* C-MPNN without readout (denoted as C-MPNN-*basic* in Table 9), all hyper-parameters used are reported in Table 11, and we adopt layer-normalization [2] and short-cut connection after each aggregation and before applying ReLU. We also discard the edges that directly connect query node pairs to prevent overfitting. For RGCN, we use two layers, each with a hidden dimension of 100. We consider the same basis-decomposition trick with a basis number equal to 30. In addition, we train the model with a learning rate of 0.1 and a dropout rate of 0.2, using Adam optimizer for 10,000 epochs. For CompGCN, we use three layers, each with hidden dimension 200, and adopt *distmult* scoring function as well as element-wise multiplication as a composition function. During the training process, we consider a learning rate of 0.001 and a dropout rate of 0.1, using Adam optimizer for 500 epochs. Additionally, since there are no node features for both of the datasets, we initialize node representations using learnable embeddings with Xavier initialization for RGCN and CompGCN. We take the results of NeuralLP from Yang et al. [35], and the results of DRUM from Sadeghian et al. [25], respectively. The best checkpoint for each model instance is selected based on its performance on the validation set. All experiments are performed on one NVIDIA V100 32GB GPU.

**Evaluation.** We consider *filtered ranking protocol* [6] with 1 negative sample per positive triplet, and report Mean Rank(MR), Mean Reciprocal Rank(MRR), and Hits@10 for each model. We also report averaged results of *five* independent runs for all experiments.

**Results.** From Table 9, it is evident that models under the class of C-MPNN consistently outperform R-MPNN models across both datasets and all evaluation metrics, with only one exception on FB15k-237 with NeuralLP. This aligns with our theoretical understanding, as C-MPNN models are inherently more expressive than R-MPNNs. In particular, the C-MPNN-*basic* model stands out by achieving the lowest MR and the highest values for both MRR and Hits@10, surpassing all other models by a significant margin, underscoring its efficiency and robustness in transductive knowledge graph completion tasks.

## D.2 Experiments on biomedical datasets

To compare R-MPNNs and C-MPNNs on large-scale graphs, we carried out additional transductive knowledge graph completion experiments on biomedical datasets: Hetionet [15] and ogbl-biokg [16].

**Datasets.** Hetionet [15] is a large biomedical knowledge graph that integrates data from 29 different public databases, representing various biological connections and associations, and ogbl-biokg is a large-scale biomedical knowledge graph dataset, developed as a part of the Open Graph Benchmark (OGB) [16] suite. We have used the provided standardized train-test split for both datasets and similarly augmented every fact $r(u, v)$ with its inverse relation $r^{-1}(u, v)$. We present the detailed dataset statistics in Table 13.

**Implementation.** We study the *basic* C-MPNN without readout, referred to as C-MPNN in the Table 12, the epoch number is set to 1 due to the datasets being larger and denser, leading to longer execution times for C-MPNN. All hyperparameters used for C-MPNN are reported in Table 14, and we adopt layer-normalization [2] and short-cut connection after each aggregation and before applying ReLU. We also discard the edges that directly connect query node pairs. For RGCN, we consider a batch size of 65536, a learning rate of 0.001, and 100 epochs for both datasets. On Hetionet, we use a hidden dimension size of 200, and a learning rate of 0.001 across 4 layers. On ogbl-biokg, we consider a dimension size of 500, and a dropout rate of 0.2 across 2 layers. For CompGCN on Hetionet, the parameters include a batch size of 65536, 100 epochs, a dimension size of 200, a learning rate of 0.01, and a single layer. We adopt *distmult* scoring function and element-wise multiplication as a composition function. On ogbl-biokg, CompGCN results cannot be reproduced to the best of our effort due to an out-of-memory (OOM) error. The best checkpoint for each model instance is selected based on its performance on the validation set. All experiments are performed on one NVIDIA V100 32GB GPU. In addition, since there are no node features for both of the datasets, we initialize node representations using learnable embeddings with Xavier initialization for RGCN and CompGCN.

**Evaluation.** We consider *filtered ranking protocol* [6], but restrict our ranking to entities of the same type. With each positive triplet, we use 32 number of negative samples and report the averaged results of Mean Reciprocal Rank(MRR), Hits@1, and Hits@10 over *five* independent runs for each model.

**Results.** From Table 12, it is evident that C-MPNN outperforms R-MPNN on both datasets by a large margin, despite the challenges posed by the large and dense nature of biomedical knowledge graphs. These results are reassuring as they have further consolidated our theory, showing that the expressivity gain of C-MPNNs compared to R-MPNNs has a significant impact on real-world biomedical datasets.

Table 12: Transductive experiments on biomedical knowledge graphs. We use *basic* C-MPNN architecture with $\text{AGG} = \text{sum}$, $\text{MSG} = \text{MSG}_r^1$, and $\text{INIT} = \text{INIT}^2$ with no readout component. OOM stands for out of memory.

| Model | Hetionet | | | ogbl-biokg | | |
|---|---|---|---|---|---|---|
| | **MRR** | **Hits@1** | **Hits@10** | **MRR** | **Hits@1** | **Hits@10** |
| RGCN | 0.120 | 0.067 | 0.228 | 0.636 | 0.511 | 0.884 |
| CompGCN | 0.152 | 0.083 | 0.292 | OOM | OOM | OOM |
| C-MPNN | **0.479** | **0.394** | **0.649** | **0.790** | **0.718** | **0.927** |

Table 13: Dataset statistics for transductive biomedical knowledge graph completion.

| Dataset | #Nodes | #Node Type | #Relation | #Triplet #Train | #Valid | #Test |
|---|---|---|---|---|---|---|
| Hetionet | 47,031 | 11 | 24 | 1,800,157 | 225,020 | 225,020 |
| ogbl-biokg | 93,773 | 5 | 51 | 4,762,677 | 162,886 | 162,870 |

Table 14: Hyperparameters for the transductive experiments on biomedical knowledge graphs with C-MPNN.

| Hyperparameter | | Hetionet | ogbl-biokg |
|---|---|---|---|
| **GNN Layer** | Depth($T$) | 4 | 6 |
| | Hidden Dimension | 32 | 32 |
| **Decoder Layer** | Depth | 2 | 2 |
| | Hidden Dimension | 32 | 32 |
| **Optimization** | Optimizer | Adam | Adam |
| | Learning Rate | 2e-3 | 2e-4 |
| **Learning** | Batch size | 64 | 8 |
| | #Negative Sample | 32 | 32 |
| | Epoch | 1 | 1 |
| | Adversarial Temperature | 1 | 0.5 |

## E  Evaluating the power of global readout on a synthetic dataset

We constructed a synthetic experiment to showcase the power of global readout (**Q5**).

**Dataset.** We proposed a synthetic dataset TRI-SQR, which consists of multiple pairs of knowledge graphs in form $(G_1, G_2)$ s.t. $G_1 = (V_1, E_1, R, c_1)$, $G_2 = (V_2, E_2, R, c_2)$ where $R = \{r_0, r_1, r_2\}$. For each pair, we constructed as follows:

1. Generate an Erdos-Renyi graph $G_{\text{init}}$ with 5 nodes and a random probability $p$. We randomly select one of the nodes as the source node $u$.

2. $G_1$ is constructed by disjoint union $G_{\text{init}}$ with two triangles, one with edges relation of $r_1$, and the other with $r_2$. The target query is $r_3(u, v)$ for all $v$ in a triangle with an edge relation of $r_1$.

3. Similarly, $G_2$ is constructed by disjoint union another copy of $G_{\text{init}}$ with two squares, one with edges relation of $r_1$, and the other with $r_2$. The target query is $r_3(u, v)$ for all $v$ in a square with an edge relation of $r_2$.

One example of such a pair can be shown in Figure 6. We generate 100 graph pairs and assign 70 pairs as the training set and the remaining 30 pairs as the testing set (140 training graphs and 60 testing graphs). In total, there are 490 training triplets and 210 testing triplets.

**Objective.** For all node $v \notin G_{\text{init}}$, we want a higher score for all the links $r_0(u, v)$ if either $v$ is in a triangle consists of $r_1$ relation or $v$ is in a square consists of $r_2$ relation, and a lower score otherwise. For negative sampling, we choose counterpart triplets for each graph, that is, we take $r_0(u, v)$ for all $v$ in a triangle with an edge relation of $r_2$ in $G_1$ and in a square with an edge relation of $r_1$ in $G_2$.

**Model architectures.** We have considered two model architectures, namely C-MPNNs with $\text{INIT} = \text{INIT}^2$, $\text{AGG} = \text{sum}$, $\text{MSG} = \text{MSG}_r^2$, and $f(t) = t$ without sum global readout:

$$\boldsymbol{h}_{v|u,q}^{(0)} = \mathbb{1}_{u=v} * \mathbf{z}_q,$$

$$\boldsymbol{h}_{v|u,q}^{(t+1)} = \sigma\Big(\mathbf{W}_0^{(t)}\big(\mathbf{h}_{v|u,q}^{(t)} + \sum_{r \in R} \sum_{w \in \mathcal{N}_r(v)} \text{MSG}_r^2(\mathbf{h}_{w|u,q}^{(t)}, \mathbf{z}_q)\big)\Big),$$

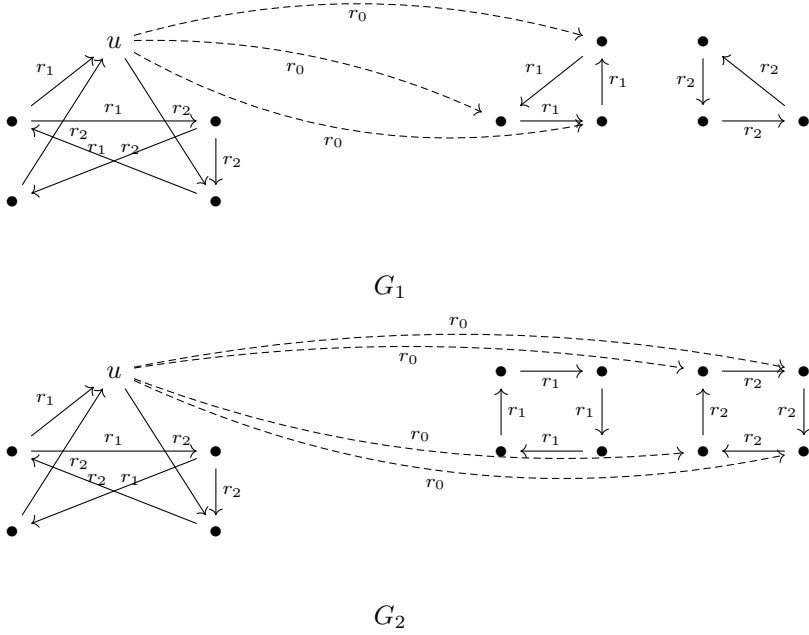

$$G_1$$

$$G_2$$

Figure 6: Construction of graph pair in TRI-SQR for global readout

and the same C-MPNN architecture with sum global readout:

$$\boldsymbol{h}_{v|u,q}^{(0)} = \mathbb{1}_{u=v} * \mathbf{z}_q,$$

$$\boldsymbol{h}_{v|u,q}^{(t+1)} = \sigma\Big(\mathbf{W}_0^{(t)}\big(\mathbf{h}_{v|u,q}^{(t)} + \sum_{r \in R}\sum_{w \in \mathcal{N}_r(v)} \text{MSG}_r^2(\mathbf{h}_{w|u,q}^{(t)}, \mathbf{z}_q)\big) + \mathbf{W}_1^{(t)}\sum_{w \in V} \boldsymbol{h}_{w|u,q}^{(t)}\Big),$$

**Design.** We claim that C-MPNNs with sum readout can correctly predict all testing triplets, whereas C-MPNNs without sum readout will fail to learn this pattern and achieve $50\%$ as random guessing. Theoretically, the TRI-SQR dataset is designed in such a way that any R-MPNN model assigns identical node representations to nodes in triangles and squares with the same relation type. Consequently, any C-MPNN model without global readout will be unable to determine which graph between $G_1$ and $G_2$ in the graph pair is being predicted, making it challenging to learn the conditional representation $\boldsymbol{h}_{u|v,q}$. However, we anticipate that a C-MPNN model with sum readout can differentiate between $G_1$ and $G_2$ in each pair, as it can access global information like the total number of nodes in the graph. This allows it to accurately identify the graph being predicted in the graph pair, even when the representations of the triangle and square with the same relation are identical. As a result, it can learn the target rules and achieve $100\%$ accuracy.

**Experiment details.** We configure each model variant with four layers and 32 hidden dimensions per layer. We set a learning rate of 1e-4 for both models and train them for 500 epochs. Empirically, we find that C-MPNN with global sum readout achieves $100\%$ accuracy, while C-MPNN without global sum readout reaches $50\%$ accuracy as random guesses, which is consistent with our theoretical expectations.

