# OpenReview forum: "A Theory of Link Prediction via Relational Weisfeiler-Leman on Knowledge Graphs"
_NeurIPS.cc/2023/Conference — NeurIPS 2023 poster_

### Official Review · Reviewer_1Kyu · 2023-06-17

**Soundness:** 3 good
**Presentation:** 2 fair
**Contribution:** 3 good
**Rating:** 6
**Confidence:** 2

**Summary:**

This paper focuses on graph neural networks (GNNs) for link prediction on knowledge graphs. It introduces the concept of conditional message-passing neural networks, which compute pairwise node representations based on a source node and a query relation. The paper analyzes the expressive power of these networks with a relational Weisfeiler-Leman algorithm and a logical characterization. Experimental analysis is conducted to validate the theoretical findings.


**Strengths:**

1. The paper offers a systematic understanding of conditional message-passing neural networks for link prediction on knowledge graphs.

2. The paper provides insights into the expressive power of conditional message-passing neural networks over the general message-passing neural networks.

3. The paper conducts experimental analysis to empirically validate the theoretical findings. It explores the impact of different model choices, e.g., initialization, global readout functions, providing practical insights into the performance of graph neural networks for link prediction on knowledge graphs.

**Weaknesses:**

1.  The title and some lines in the introduction seem to be a little bit overclaimed. For instance, the title named a theory of link prediction seems to be claimed on the general link prediction task while the main discussion in the main paper is about the Knowledge Graph link prediction. besides, it is hard for me to understand lines 32-36. I could see from this paragraph that the paper focuses on the NBFNet-based model with conditional message passing. Nonetheless, it seems not so clear to me what higher-order messaging passing neural network computing the pairwise representation on the node. I would like to suggest authors to strengthen the focus of the paper at the beginning and then detail the difference between the RGCN, and CompGCN models with NBFNet.
2.  The second bullet of the contribution part at line 52 claims that the paper designed a relational Weisfeiler-Lemma algorithm. It leads to the potential misleading that the paper is the first to propose the relational Weisfeiler-Lemma algorithm. Nonetheless, as discussed in line 97 in related work, the relationship is already included in the discussion in 'Weisfeiler and Leman go relational'. I would like to suggest the authors for a clear comparison with the related work on existing relational Weisfeiler-Lemma algorithms.
3.  The intuition for logical characterization is not so clear. It seems to be hard for me to understand the logical characterization. How it contributes to the expressiveness is not so clear. I would like to suggest the author have more discussion on motivation and why the contribution is significant.
4.  Lack of explanation on why only focuses on the inductive Link Prediction setting. Despite the NBFNet shows strong ability typically inductive setting. Nonetheless, the algorithm seems not so relevant in the transductive or inductive setting. I still think the experiments on both transductive and inductive settings should be taken into consideration.

**Questions:**

See in Weakness

**Limitations:**

The authors have adequately addressed the limitations

---

> ### Author Rebuttal · Authors · 2023-08-09
>
> Thank you for your review and your comments on our paper. We address each below.
>
> >“The title and some lines in the introduction seem to be a little bit overclaimed. For instance, the title named a theory of link prediction seems to be claimed on the general link prediction task while the main discussion in the main paper is about the Knowledge Graph link prediction.”
>
> We do not mean to overclaim our findings: we state link prediction in the title, because link prediction on simple graphs can be seen as a special case of link prediction on KGs, where only one relation type is allowed. In fact, all of the studied models and the presented theory equally applies to link prediction on single-relational graphs. The empirical focus is on KGs, but we make this clear in the paper. If there is an ambiguity regarding this, we will do our best to tone things accordingly to avoid any confusion.
>
> >“Besides, it is hard for me to understand lines 32-36. I could see from this paragraph that the paper focuses on the NBFNet-based model with conditional message passing. Nonetheless, it seems not so clear to me what higher-order messaging passing neural network computing the pairwise representation on the node.”
>
>
> This statement refers to the following: In principle, one could design a relational variant of  higher-order GNNs, especially those with expressive power matching that of folklore 2-WL. These models compute strong pairwise node invariants, which is crucial for link prediction. However, they are inherently unscalable, which hinders their use in practice. This is precisely what motivates a trade-off between the expressive power and scalability: our paper studies models which are scalable while also being able to compute pairwise node invariants. We can improve this explanation also in the paper.
>
> >“The second bullet of the contribution part at line 52 claims that the paper designed a relational Weisfeiler-Lemma algorithm. It leads to the potential misleading that the paper is the first to propose the relational Weisfeiler-Lemma algorithm. Nonetheless, as discussed in line 97 in related work, the relationship is already included in the discussion in 'Weisfeiler and Leman go relational'.”
>
> We will rephrase this sentence to avoid any misleading statements as:
>
> *“We define a relational Weisfeiler-Leman algorithm (building on similar works such as Barceló et al [1]), and prove that conditional message passing neural networks can match the expressive power of this algorithm.”*
>
> As you kindly acknowledge, we give appropriate credit to this work in multiple places in the paper. Specifically, we cover this in the related work section, and separately, on line 484 as *“...generalizes results from Barceló et al [1]”*, and finally, in Appendix A.1 as: *“The expressive power of R-MPNNs has been recently characterized in terms of a relational variant of the Weisfeiler–Leman test [1]”*.
>
> >“The intuition for logical characterization is not so clear.…How it contributes to the expressiveness is not so clear.”
>
> Logical characterizations are by now a well-established topic related to the expressiveness of GNNs (starting from the work of Barceló et al [2]). Unlike the characterization of GNNs based on the WL test, these logical characterizations are *uniform*, in the sense that they hold over the set of all graphs. This means that for any unary formula F(x) in the logical language there is a GNN A such that, *over every graph G*, the set of nodes from G that satisfy F(x) is the same as the set of nodes from G that are classified positively by A. We believe that this logical characterization provides a clear picture of what C-MPNNs can do. On the one hand, the logic is “guarded”, i.e., it is only possible to existentially quantify over nodes that are directly linked to our given node x. This represents the “local” nature of C-MPNNs. On the other hand, the logic has counting quantifiers, which represents the fact that C-MPNNs can distinguish how many neighbors of x satisfy a certain property.
>
> Logical characterizations are also important since they provide a link between the procedural behavior of GNNs and the more declarative and better understood formalisms of logic. For example, and as established by Barcelo et al [2], *“if one proves that two GNN architectures are captured with two logics, then one can immediately transfer all the knowledge about the relationships between those logics, such as equivalence or incomparability of expressiveness, to the GNN setting.”* As an application of this, we have shown in the paper that C-MPNNs with global readout are strictly more powerful than standard C-MPNNs for expressing logical classifiers: we have done so by showing that there is a logical classifier that the former can recognize but the latter cannot.
>
> >“Lack of explanation on why only focuses on the inductive Link Prediction setting. Despite the NBFNet shows strong ability typically inductive setting. Nonetheless, the algorithm seems not so relevant in the transductive or inductive setting. I still think the experiments on both transductive and inductive settings should be taken into consideration.”
>
> Yes, the proposed algorithms are not limited to the inductive setting. As a matter of fact, we did take this into account in our paper, and also reported transductive link prediction experiments in Appendix B.3 of our original submission.
>
> Following the suggestions, we further experimented on transductive link prediction on biomedical knowledge graphs, **Hetionet** and **OGB-Biokg** which is reported in Table 3 of **rebuttal.pdf**. These empirical results are very much in line with the presented theory.
>
> *[1] Pablo Barceló, Mikhail Galkin, Christopher Morris, and Miguel Romero. Weisfeiler and leman go relational. LoG, 2022.*
>
> *[2] Pablo Barceló, Egor V. Kostylev, Mikaël Monet, Jorge Pérez, Juan L. Reutter, and Juan Pablo Silva. The logical expressiveness of graph neural networks., ICLR, 2020.*

---

> > ### Comment · Reviewer_1Kyu · 2023-08-10
> > **Response**
> >
> > Thanks for your response. I still think Link Prediction and KGC are different problems, and they should not discuss together, not from a theoretical perspective, but from a practical perspective. I would raise my score if this W1 is solved

---

> > > ### Author Response · Authors · 2023-08-10
> > > **Adjusting the title**
> > >
> > > Thanks for your comment! We agree with your comment regarding the practical aspect. We acknowledged in our response that the empirical focus is on KGs. To make this also explicit in the title, we propose the following title: A Theory of Link Prediction via Relational Weisfeiler-Leman on Knowledge Graphs. We hope this addresses your concern and clears up the ambiguity. We also hope that the other concerns are addressed in our rebuttal.

---

> > > > ### Author Response · Authors · 2023-08-14
> > > > **Follow up**
> > > >
> > > > To follow up on this, could you please let us know whether the proposed change in the title addresses your concern (W1)? We will further highlight that the focus of this work is on KGs.

---

> > > > > ### Comment · Reviewer_1Kyu · 2023-08-17
> > > > > **Response**
> > > > >
> > > > > Yes. I will raise my score, However actually I think there is still a bit of limited contribution, then I will also provide lower confidence.

---

> > > > > > ### Author Response · Authors · 2023-08-17
> > > > > >
> > > > > > Thanks for your response, and for raising your score! Our fundamental contribution is to theoretically ground the behaviour of a class of architectures, and, as a result, to concretely explain their superior empirical performance based on formal criteria. By this perspective, our study forms a formal basis for future work which aims to design new architectures at the the trade-off between expressive power and scalability. We will integrate the discussions and the findings from the rebuttal period to the new version of the paper.

---

### Official Review · Reviewer_7NY6 · 2023-07-04

**Soundness:** 4 excellent
**Presentation:** 4 excellent
**Contribution:** 3 good
**Rating:** 7
**Confidence:** 4

**Summary:**

The paper investigates graph neural networks (GNNs) for link prediction in knowledge graphs. The authors propose a GNN that generalizes several existing techniques based on the labeling trick and NBFNets and investigate its expressive power by relating it to the Weisfeiler-Leman method and giving a logical characterization. Different design choices of the architecture are explored experimentally.

**Strengths:**

* A highly relevant problem with various applications is addressed and neural architectures for related tasks are thoroughly analyzed, giving new theoretical insights. As the number of papers published in this area grows rapidly, it is of utmost importance to explain the advantage of practical design choices and to work out the benefits of specific architectures. The paper is a solid contribution in this direction.
* The authors give a good overview of related work, including several recent results, and place their work well in the context of existing approaches. The paper continues recent work and generalizes it, which is a valuable contribution to fostering future work.
* The paper is well-written and enjoyable to read; the presentation is on a high level.

**Weaknesses:**

* The experimental comparison focuses on the design choices of the proposed architecture but does not compare to existing works. This would be desirable, in particular, since higher-order GNNs subsume some of the techniques (as stated in the paper). The trade-off between efficiency and expressivity appears to be crucial but is neither investigated theoretically in terms of time complexity nor experimentally in terms of measured runtimes.
* The presentation of conditional messages passing remains abstract. It would be helpful to come up with an illustrative example, if possible.

Minor comments:
* l176: $\epsilon_u$ should be explained
* l186: $\mathbf{z}_q$ as argument of MSG should be just $q$ following the notation introduced earlier
* l370: "Thus, thus"

**Questions:**

* How does the approach compare to NBFNets, architecture using the labeling trick, and higher-order WL-type GNNs regarding (i) accuracy and (ii) running time?


**Limitations:**

Limitations are addressed sufficiently.

---

> ### Author Rebuttal · Authors · 2023-08-09
>
> Thank you for your review and your comments on our paper. We address each below.
>
> >“The experimental comparison focuses on the design choices of the proposed architecture but does not compare to existing works. This would be desirable, in particular, since higher-order GNNs subsume some of the techniques (as stated in the paper). The trade-off between efficiency and expressivity appears to be crucial but is neither investigated theoretically in terms of time complexity nor experimentally in terms of measured runtimes.”
>
> Thanks for raising this. Following your suggestion, we present a time-complexity analysis for C-MPNNs and other inductive link prediction models in Table 4 of **rebuttal.pdf**. This also includes the amortized time per query which can better reflect the practical use cases and the advantages stemming from parallelizability. Regarding higher-order GNNs we consider the neural architecture from Barceló et al. [1] corresponding to rwl$_2$ (and mentioned in line 259 of our paper). This higher-order model requires $O(|V||E|d + |V|^2d^2)$ time in each layer to update all pairwise representations, and it cannot be parallelized, hence remains prohibitive in practice.
>
> >“The presentation of conditional messages passing remains abstract. It would be helpful to come up with an illustrative example, if possible.”
>
> The main difference from standard message passing stems from the conditioning on the source node, which we make explicit in our approach. We agree this is more intricate than standard message passing, and hence we provide a visualization of C-MPNN and R-MPNN in Figure 1 of **Rebuttal.pdf** with the following explanation: for the basic C-MPNN model, in order to compute a query fact $q(u,v)$, where $u$ is the source node and $v$ is the target node, we first initialize all node representations with the zero vector except the representation of the node $u$ which is assigned a learnable query vector $\mathbf{z}_q$. Following the initialization, we carry out relational message passing and decode the hidden state of the target node $v$ to obtain the output, which yields the representation of $v$ conditioned on $u$.
>
>
> >“Minor comments: l176: $\epsilon_u$ should be explained. ”
>
> $\epsilon_u$ is explained in line 184: *“adding an error vector $\epsilon_{u}$ sampled from $\mathcal{N}(0,1)$ to the conditioned node's initialization”*. Here, $\epsilon_{u}$ is a fixed perturbation given that node $u$ is chosen.
>
> >“Minor comments: …l186: $\mathbf{z}_q$ as argument of MSG should be just $q$ following the notation introduced earlier. ”
>
> Many thanks for this! Indeed, this is a slight abuse of notation. We will pass $\mathbf{z}_q$ as an argument to MSG on line 150, where this is first introduced to fix this.
>
> >“Minor comments: …l370: "Thus, thus".”
>
> Thank you, we have corrected this.
>
> >“How does the approach compare to NBFNets, architecture using the labeling trick, and higher-order WL-type GNNs regarding (i) accuracy and (ii) running time?”
>
> We answer your questions regarding (i) accuracy and (ii) runtime, separately for each model:
> - **Comparison with NBFNets**: Let us note that C-MPNNs subsume NBFNets. Indeed, if we consider C-MPNNs without readout and set the history function as $f(t) = 0$, we obtain (a slight generalization of) NBFNets, as stated in the paper. In terms of the runtime, C-MPNNs and NBFNets are comparable: even though the readout component of C-MPNNs incurs a linear overhead, this is dominated by other factors. In terms of the accuracy, the picture is more complex: our results show that the choice of history function is not critical, which is empirically confirmed: $f(t) = t$ and $f(t) = 0$ yields similar results (Table 1 of the main paper). On the other hand, the (uniform) logical characterization shows the theoretical benefit of using a global readout component: this is empirically observed on FB15K-237, where the model  using a readout component showed substantial gains (leading to state-of-the-art results) compared to not using a readout component (Table 2 of the main paper).
> - **Comparison with the architectures using labelling trick**: GraIL is an architecture using labeling trick and we focus on this to make the comparison concrete. In terms of runtime, architectures which rely on labeling trick are typically slower, since they need to label both the source and the target node, as we state in the paper *“... the Labeling Trick only applies when both the source $u$ and target nodes $v$ are labeled”*. This yields worse runtime complexity for these models, and particularly for GraIL: we report the runtimes in Table 3 of **rebuttal.pdf**. In C-MPNNs, we only label source $u$, which allows parallel comparison for all queries in form $q(u,?)$: a single forward pass would compute all hidden representations of these queries. With Labeling Trick, $|V|$ forward passes need to be carried out as each time, both source $u$ and target $v$ needs to be specified. In terms of accuracy, we added more baselines in **rebuttal.pdf**, including GraIL, showing C-MPNNs outperform these models. We will integrate these discussions to our paper.
> - **Comparison with higher-order GNNs**: Regarding higher-order GNNs we consider the neural architecture from Barceló et al. [1] corresponding to rwl$_2$, as we stated in a response to a related question. These higher-order models are computationally prohibitive in practice, and hence it is hard for us to conclude anything in terms of the accuracy of these models on knowledge graphs. Theoretically, they can compute stronger binary invariants, and hence we conjecture that they would perform strongly if it would have been possible to scale them up. The trade-off between expressive power and scalability is perhaps the most intriguing aspect of the study of link prediction.
>
>
> *[1] Pablo Barceló, Mikhail Galkin, Christopher Morris, and Miguel Romero. Weisfeiler and leman go relational. LoG, 2022.*

---

> > ### Comment · Reviewer_7NY6 · 2023-08-16
> >
> > Thank you for your detailed reply. My questions regarding the trade-off between expressivity and complexity have been addressed sufficiently.

---

> > > ### Author Response · Authors · 2023-08-17
> > >
> > > Thank you very much for going through our rebuttal, and for your continued positive evaluation of our paper.

---

### Official Review · Reviewer_kRBD · 2023-07-05

**Soundness:** 4 excellent
**Presentation:** 3 good
**Contribution:** 3 good
**Rating:** 5
**Confidence:** 3

**Summary:**

The authors note that while GNNs are understood well in the context of simple graphs, there is a lack of comprehensive understanding when it comes to knowledge graphs. This study aims to systematically explore the use of GNNs in knowledge graphs, specifically in relation to the task of link prediction. The research includes a unifying view on different, seemingly unrelated models, potentially revealing new ones. The study evaluates the expressive power of different models using the relational Weisfeiler-Leman algorithm. The theoretical findings help clarify the advantages of some commonly used practical design choices, and these theories are supported by empirical validation.

**Strengths:**

I have read the appendix of the paper. The technical details are clearly articulated, and the proof statements are clear and persuasive.

The authors introduce conditional message-passing neural networks and define a relational Weisfeiler-Leman algorithm to substantiate that these neural networks can match the expressive power of this algorithm.

The initialization progress is interesting.


**Weaknesses:**

In the background section, the author first mentioned G = (V, E, R, c) while change it to G = (V, E, R, x) later. I know the author says it usually should be x instead of c. But can we directly use G = (V, E, R, x) at first to prevent confusion?

More GNN baseline method should be involved in the experiments.

The proof process is commendable, but could it be possible to draw a conclusion from it and incorporate it into the main text instead of the appendix?


**Questions:**

Can the author add more baselines?

Although the authors have already provided two datasets, each with four different versions, I think this is adequate, so I have not listed it as a weakness. However, it would be good if the authors could include one or two additional datasets.


**Limitations:**

As the author mentioned, the method is limited to binary task such as link prediction.

---

> ### Author Rebuttal · Authors · 2023-08-09
>
> Thank you for your review and your comments on our paper. We address each below.
>
> >“In the background section, the author first mentioned $G = (V, E, R, c)$ while change it to $G = (V, E, R, \mathbf{x})$ later. I know the author says it usually should be $\mathbf{x}$ instead of $c$. But can we directly use $G = (V, E, R, \mathbf{x})$ at first to prevent confusion?”
>
> We understand the source of the confusion, but let us note that $G = (V, E, R, c)$ is indeed a more general definition compared to $G = (V, E, R, \mathbf{x})$. This is indicated in the background section as follows: *“When $D=\mathbb{R}^d$, we also say that $c$ is a $d$-dimensional *feature map*, and typically use $\mathbf{x}$ instead of $c$”*. The differences between $G = (V, E, R, c)$ and $G = (V, E, R, \mathbf{x})$ may appear rather subtle but they are important. In our context, we use $c$ to denote a discrete set of colors used in the WL test, whereas $\mathbf{x}$ are the continuous features belonging to $\mathbb{R}^d$ in the corresponding neural counterpart. We carefully distinguish between these two notations on knowledge graphs for mathematical clarity as the lack of such a notational convention can present problems. We will highlight this point to avoid the confusion of readers.
>
> >“More GNN baseline method should be involved in the experiments.”
>
> We added more baseline methods from GraIL paper [1] which are reported in Table 1 of the **rebuttal.pdf** as part of our global response. We also provided the transductive link prediction baseline methods from DRUM paper [2], which are reported in Table 2 of the **rebuttal.pdf**.
>
> >“The proof process is commendable, but could it be possible to draw a conclusion from it and incorporate it into the main text instead of the appendix?”
>
> Thanks for going through the proofs. We are glad you find the presentation of the technical content commendable. The key theoretical contributions are presented in terms of Theorem 5.1, 5.2, and 5.3 along with brief explanations. Figure 1 serves as a summary of all the lemmas presented in Section 5.3. We can elaborate these details further (with more space allowance): we are happy to add some discussions regarding the high-level ideas behind these proofs in the body of the paper.
>
> >“Although the authors have already provided two datasets, each with four different versions, I think this is adequate, so I have not listed it as a weakness. However, it would be good if the authors could include one or two additional datasets.”
>
> Thanks for this suggestion! We agree that diverse datasets are beneficial to strengthen our point. Following your suggestion, we provide transductive link prediction experiments on two additional datasets: **Hetionet** and **OGB-Biokg**. The experiments are reported in Table 3 of **rebuttal.pdf** as part of our global response. The results are reassuring as they show very similar trends to those observed on the other datasets. Please let us know whether these points clarify your concerns.
>
> *[1] Komal Teru, Etienne Denis, and Will Hamilton. Inductive relation prediction by subgraph reasoning. ICML, 2020.*
>
> *[2] Ali Sadeghian, Mohammadreza Armandpour, Patrick Ding, and Daisy Zhe Wang. Drum: End-to-end differentiable rule mining on knowledge graphs. NeurIPS, 2019.*

---

> > ### Comment · Reviewer_kRBD · 2023-08-18
> >
> > Thank you for your response. You address some of my concerns, I'd like to raise my score.

---

> > > ### Author Response · Authors · 2023-08-18
> > >
> > > Thank you very much for going through the rebuttal, and for raising your score. Could you please let us know if there are any remaining concerns? This would allow us to hopefully clarify these concerns and get your final feedback before the rebuttal window closes. To the best our understanding, your main concern was regarding the lack of a high-level presentation of the proof sketches in the body of the paper: we are happy to integrate these explanations to the main paper to present our results at different granularities so that the reader gets a better idea of the proofs without going through them in detail. We hope this suggested change is satisfactory and fully addresses your concerns. Of course, we are happy to elaborate based on your input.

---

> > > > ### Comment · Reviewer_kRBD · 2023-08-19
> > > >
> > > > Thank you for your response. I understand your revision and recognize that there is a trade-off between detailed explanation and a summarized main idea. However, I believe that the current representation might still confuse users who do not read the detailed proof. Therefore, I would like to maintain my current score.

---

> > > > > ### Author Response · Authors · 2023-08-20
> > > > >
> > > > > Thanks for acknowledging this trade-off which is almost always present in theoretical papers. To better inform you on how the high-level explanations would look like, we include these explanations for the sake of completeness:
> > > > >
> > > > > - __Overview of Theorem 5.1:__ We first show a correspondent characterization of the expressive power of R-MPNNs in terms of a relational variant of the WL test (Theorem A.1). This result generalizes results from Barceló et al [1]. Then we apply a reduction from C-MPNNs to R-MPNNs, that is, we carefully build an auxiliary knowledge graph (encoding the pairs of nodes of the original knowledge graph) to transfer our R-MPNN characterization to our sought C-MPNN characterization.
> > > > >
> > > > > - __Overview of Theorem 5.2:__ We start by showing a logical characterization for R-MPNNs (without global readout) in terms of a variant of graded modal logic called rGFO$^2_{\text{cnt}}$ (Theorem A.11), which generalizes results from Barceló et al [2] to the case of multiple relations. Then, as in the case of Theorem 5.1, we apply a reduction from C-MPNNs to R-MPNNs (without global readout) using an auxiliary knowledge graph and a useful translation between the logics rGFO$^2_{\text{cnt}}$ and rGFO$^3_{\text{cnt}}$.
> > > > >
> > > > > - __Overview of Theorem 5.3:__ Intuitively, our logic rGFO$^3_{\text{cnt}}$ from Theorem 5.2 only allows us to navigate the graph by moving to neighbors of the “current node”. The logic erGFO$^3_{\text{cnt}}$ is a simple extension that allows us to move also to non-neighbors (Proposition A.15 shows that this logic actually gives us more power). Adapting the translation from logic to GNNs from Theorem A.11, we can easily show that C-MPNNs with global readout can capture this extended logic.
> > > > >
> > > > > _[1] Pablo Barceló, Mikhail Galkin, Christopher Morris, and Miguel Romero. Weisfeiler and leman go relational. LoG, 2022._
> > > > >
> > > > > _[2] Pablo Barceló, Egor V. Kostylev, Mikaël Monet, Jorge Pérez, Juan L. Reutter, and Juan Pablo Silva. The logical expressiveness of graph neural networks., ICLR, 2020._

---

### Official Review · Reviewer_cjW9 · 2023-07-31

**Soundness:** 3 good
**Presentation:** 3 good
**Contribution:** 2 fair
**Rating:** 4
**Confidence:** 3

**Summary:**

This paper explores the expressive power of several GNNs designed for link prediction in knowledge graphs. The authors propose a conditional message passing framework with various designs for each component, a generalization of NBFNets. They also prove that the proposed framework can match the expressive power of a relational Weisfeiler-Leman algorithm. The experimental results demonstrate the impact of different model choices.

**Strengths:**

1. The paper provides a theoretical understanding of some relational GNNs for knowledge graphs, such as NBFNets.
2. The proposed C-MPNN is reasonable, generalizing GNNs that can compute pairwise representations.
3. The authors study the effect of different model architectures, adding depth to the exploration of GNNs.

**Weaknesses:**

1. Using relational asymmetric local 2-WL (rawl2) to measure the expressive power of C-MPNN is a little awkward. The definition of rawl2 is quite similar to C-MPNN, leading to potential confusion.
2. The experiments do not verify some theoretical findings, focusing mainly on different choices of model design. Certain theoretical discoveries, such as logical characterization, are not confirmed.
3. The experiments are primarily conducted on two datasets with differing data splits, leading to inconsistencies. More datasets and deeper analysis may be necessary to solidify the findings.

**Questions:**

Please refer to the weakness.

---

> ### Author Rebuttal · Authors · 2023-08-09
>
> Thank you for your review and your comments on our paper. We address each below.
>
> >“Using relational asymmetric local 2-WL (rawl$_2$) to measure the expressive power of C-MPNN is a little awkward. The definition of rawl$_2$ is quite similar to C-MPNN, leading to potential confusion.”
>
> The color refinement algorithm rawl$_2$ is introduced to exactly capture the theoretical expressiveness of the corresponding neural architecture C-MPNNs. These algorithms are therefore closely related to each other, which can perhaps be better understood in analogy to the relation between the 1-WL algorithm and standard GNNs. Fundamentally, rawl$_2$ is a non-parameterized color refinement algorithm, whereas C-MPNNs is a class of neural networks with trainable parameters. In this sense, C-MPNNs can be viewed as a learnable, continuous, differentiable version of rawl$_2$. We are happy to elaborate further if this does not address reviewers' concerns.
>
> >“The experiments do not verify some theoretical findings, focusing mainly on different choices of model design. Certain theoretical discoveries, such as logical characterization, are not confirmed.”
>
> There are two main conclusions which follow from the presented expressiveness study and logical characterization and are empirically validated in the appendix of the paper:
>
> 1. It follows that C-MPNNs are more expressive than R-MPNNs via the abstraction given by rawl$_2$ and rwl$_1$, respectively. To empirically validate this, we conducted transductive link prediction experiments which are reported in Appendix B.3 (as part of the supplementary material). The presented results suggest a clear trend that C-MPNN has an overall better performance compared to R-MPNNs. These experiments are transductive, as R-MPNNs (i.e., R-GCN) only apply in this setup.
> 1. It follows from the logical characterization that the addition of global readout yields more expressive power in the class of queries that can be captured. We conducted a dedicated synthetic experiment to validate this, which is reported in Appendix B.1.  C-MPNN model without readout achieved random accuracy on this task, whereas the C-MPNN model with readout solved this task almost perfectly. Please note that this is in addition to the experiments on real-world data which is presented in the body of the paper.
>
> We wish to note that we prioritized inductive experiments in the body of the paper in order to validate the other aspects of the presented theory (i.e., the choice of history should not matter). We are happy to include more details regarding the experiments from the appendix if they need to be better highlighted.
>
> >“The experiments are primarily conducted on two datasets with differing data splits, leading to inconsistencies.”
>
> We are using the exact same data splits from the existing literature and our results align with the presented theory. It is therefore unclear to us where the inconsistency lies: could you please be more specific on what inconsistencies are presented in the paper so that we can clarify these?
>
> >“More datasets and deeper analysis may be necessary to solidify the findings.”
>
> We would like to re-emphasize the experiments reported in the appendix of the paper. Following your feedback, we carried out additional experiments for transductive link prediction on biomedical datasets **Hetionet** and **OGB-Biokg** to further solidify the findings on diverse datasets. These are reported in Table 3 of **rebuttal.pdf** as part of our global response. These results are reassuring, as they show very similar trends to those observed in the other datasets. Please let us know whether these points clarify your concerns.

---

> > ### Comment · Reviewer_cjW9 · 2023-08-18
> >
> > Thanks for the clarification from authors. I still have one questions:  Does logical characterization means the global readout?

---

> > > ### Author Response · Authors · 2023-08-18
> > >
> > > Thanks for going through our rebuttal! We answer your question below in two parts:
> > >
> > > ***What does the logical characterization achieve?*** When studying the expressive power of neural networks, one is typically interested in characterizing the class of functions that can be captured (or, approximated) by the neural network. In the context of graph machine learning, we are interested in the exact same question with the difference that the functions are defined over graphs. For the task of link prediction, this means the following: for each knowledge graph G, we are interested in functions of the form $f(G):  R \times V \times V \to$ {0,1} , because we want to quantify whether a link $r(a,b)$ is true or not, where $r \in R$, and $a,b \in V$.
> > >
> > > This is precisely what is achieved by logical characterizations: if we can show that a GNN architecture can capture a logic L, then it means that this GNN architecture can capture all functions which can be expressed in this logic. This is very important, because if a GNN architecture A captures a *strictly larger logic* than another GNN architecture B, then we can conclude that A is *more expressive than* B. One implication is indeed regarding the use of global readout: specifically, we obtain that C-MPNNs with global readout are strictly more powerful than standard C-MPNNs. This means that there is *at least* one function that the former can capture but the latter cannot which is empirically confirmed in Appendix B.1.
> > >
> > > ***How does this compare to WL-based characterizations?*** Logical characterizations of GNNs date back to the work of Barceló et al [1]. Unlike the characterizations of GNNs based on the WL test, these logical characterizations are uniform, in the sense that they hold over the set of all graphs. Therefore, these studies are harder to obtain and yield more powerful characterizations, because they require a single parametrized model which would apply uniformly to all graphs. Uniformity condition is much more realistic, as it applies to all graphs of all size/structure, and hence desired in theoretical analysis. Non-uniform WL characterisations are indeed weaker than logical characterisations: one example is that they cannot recognise the expressiveness gain of using of global readout which is recognised by the logical characterization, as shown in Barceló et al [1].
> > >
> > > Please let us know if this answers your question. These technical differences are intricate and easy to overlook, and we are happy to explain these more in our paper, if you agree these explanations are useful.
> > >
> > > [1] Pablo Barceló, Egor V. Kostylev, Mikaël Monet, Jorge Pérez, Juan L. Reutter, and Juan Pablo Silva. The logical expressiveness of graph neural networks., ICLR, 2020.

---

> > > > ### Author Response · Authors · 2023-08-20
> > > >
> > > > Could you please let us know whether our response answers your final question? We would also like to hear your opinion after our responses to the items 1-3 from your review. We hope your questions and concerns are adequately addressed - we are happy to elaborate more before the discussion window closes.

---

### Author Rebuttal · Authors · 2023-08-09

We thank the reviewers for their comments. We have responded to each concern in detail in our individual responses. In addition, we include a **rebuttal.pdf** to this post containing the results of all additional experiments.
Here is a summary of the changes to be made to the paper in light of the reviews received.
- **New experiments on biomedical datasets**: We added additional transductive knowledge graph completion experiments on biomedical datasets: **Hetionet** and **OGB-Biokg** and reported them in Table 3. (Reviewer cjW9, Reviewer kRBD, Reviewer 1Kyu)
- **Visualization of models**: We have provided a visualization of C-MPNN and R-MPNN in Figure 1. (Reviewer 7NY6)
- **Complexity analysis**: We presented the complexity analysis of the models in Table 4. (Reviewer 7NY6)
- **Baselines**: We added more baselines to inductive relation prediction experiments and reported these in Table 1 (Reviewer kRBD). Similarly, we added more baselines to transductive knowledge graph completion experiments and reported these in Table 2 (Reviewer kRBD).
- **Results from the appendix**: Based on reviewer feedback, we noticed that some of our experiments which are originally reported in the appendix of the paper could be better highlighted in the body of the paper. We have incorporated these changes.

We look forward to hearing your comments and feedback during the discussion period.

---

### Decision · Program_Chairs · 2023-09-21

**Decision:**

Accept (poster)

**Comment:**

The paper examines the application of Graph Neural Networks (GNNs) in predicting links within knowledge graphs. The authors introduce a novel GNN that extends various established approaches utilizing the labeling trick and NBFNets. They assess its capacity for expression by establishing connections with the Weisfeiler-Leman method and providing a logical characterization. Through experimental analysis, the paper explores different architectural options.

Overall, the paper has a solid contribution to graph neural networks. Moreover, the most of concerns raised by reviewers are adequately addressed during the rebuttal, and most of the reviewers agree to accept the paper. Thus, I also vote for acceptance.